# SPIKINGLLM: A CONVERSION-BASED METHOD WITH WINDOW INHIBITION MECHANISM FOR SPIKING LARGE LANGUAGE MODELS

## ABSTRACT

Recent advancements in large language models (LLMs) have led to unprecedented capabilities in real-world applications. However, it remains challenging to reduce the energy consumption of LLMs. In this paper, we aim to improve the energy efficiency of LLMs by leveraging the advantages of brain-inspired spiking neural networks (SNNs). We propose a novel approach called SpikingLLM, which equivalently converts quantized large language models (QLLMs) applying PrefixQuant* to their fully-spiking counterparts (all operators are in a more efficient spiking version). To ensure that every operator can be converted into its spiking version, we propose two approaches: ① QK2Head-migration post-softmax quantization, which significantly improves the performance of current QLLMs with post-softmax quantization; ② Differential-based methods, which tackle the SNN-unfriendly operators such as KV Cache. To further reduce the energy consumption, we introduce a window inhibition mechanism which effectively addresses the over-firing issue in ST-BIF$^+$ neuron and improves the sparsity. With the approaches above, SpikingLLM significantly reduces the energy consumption while achieving state-of-the-art performance on both perplexity and common-sense reasoning tasks.

## 1 INTRODUCTION

Large language models (LLMs) (Brown & Mann, 2020; Touvron & Lavril, 2023; Zhang et al., 2022; Le Scao et al., 2023) have revolutionized natural language processing (NLP) by leveraging massive-scale neural networks to achieve state-of-the-art performance across a wide range of tasks. However, the dense and continuous computations inherent in transformer-based architectures (Vaswani, 2017) pose significant challenges in terms of energy efficiency of LLMs. For instance, Llama-2-70B requires three A100-80G GPUs, each consuming approximately 400W of power (Xing et al., 2024a). These limitations are especially problematic for modern edge AI systems, which often require real-time processing under strict power constraints. To mitigate these limitations and improve the accessibility and applicability of LLMs, *we focus on energy-efficient deployment for LLMs*.

As a biologically inspired alternative to traditional artificial neural networks (ANNs), spiking neural networks (SNNs) (Maass, 1997) have emerged to bridge the gap between machine learning and neuroscience. In contrast to ANNs (LeCun et al., 2015), which rely on continuous activations, SNNs process information through discrete and event-driven spikes, closely mimicking the communication mechanisms of biological neurons (Merolla et al., 2014; Davies et al., 2018). As a result, SNNs show promising prospects on computational intelligence tasks (Roy et al., 2019) with strong autonomous learning capabilities and ultra-low power consumption (Bu et al., 2023; Ding et al., 2022; Ostojic, 2014; Zenke et al., 2015).

Unfortunately, scaling up SNNs to large-scale models remains challenging. By far, *directly training (DT)* (Zhu et al., 2023) and *ANN-to-SNN conversion (A2S)* (Xing et al., 2024a; You et al., 2024b) are two traditional methods to scale SNNs up to LLMs. *DT* unfolds the input in time-step dimension and leverages back-propagation-through-time (BPTT) (Wu et al., 2019) to update SNNs from scratch, which is computationally intensive and slow, particularly under limited computing resources. In contrast, *A2S* replaces the quantizers in quantized

ANNs (QANNs) with spiking neurons (*e.g.*, ST-BIF[+] neuron in (You et al., 2024b)), *achieving comparable performance to ANNs while significantly reducing computational costs relative to DT*. Consequently, *A2S* presents a promising pathway for scaling SNNs to LLMs. Nevertheless, applying existing *A2S* methods (You et al., 2024b; Xing et al., 2024a) directly to LLMs encounters the following challenges: ① It is challenging to construct applicable quantized LLMs (QLLMs) that ensure all operators can be converted into a spiking version, while minimizing performance degradation from quantization. ② It is difficult for *A2S* methods to establish the equivalence between QLLMs and SNNs due to the existence of SNN-unfriendly operators (*e.g.*, KV Cache, Softmax). The two challenges above are critical to convert LLMs into SNNs.

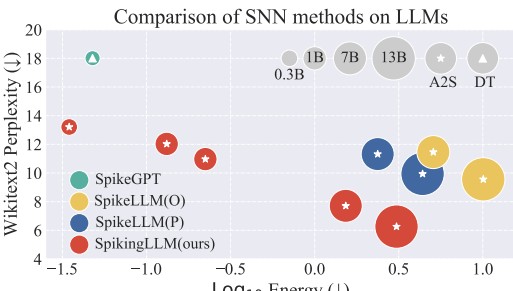

Figure 1: **Comprison of SNNs methods on LLMs**. Star and triangle marks ANN-to-SNN (A2S) and directly training (DT), respectively. SpikeLLM(O) and SpikeLLM(P) refer to SpikeLLM under OmniQuant and PrefixQuant, respectively. The area of scatter denotes model size. Results demonstrate the superiority of our SpikingLLM.

In this work, we aim to leverage the A2S method to scale SNNs up to LLMs, while maintaining all the operations in spiking version (which is defined as fully-spiking in Section 3.3). Correspondingly, we propose *SpikingLLM*, which establishes the equivalence between fully-spiking neural networks and QLLMs. SpikingLLM firstly introduces QK2Head-migration module to enable post-softmax quantization on top of PrefixQuant (Chen et al., 2025) to establish PrefixQuant[*] (in Section 4.1), ensuring all matrix products in QLLMs can be faithfully converted into their spiking versions. In addition, we refine the ST-BIF[+] neuron (You et al., 2024b) to align it with the quantizers in PrefixQuant[*] and incorporate a window inhibition mechanism, which further reduces the energy consumption. Finally, we propose SNN-friendly operators within SpikingLLM, including Spike KV Cache. Figure 1 demonstrates the superiority of our SpikingLLM over previous methods.

Our contributions are summarized as follows:

- We propose a conversion-based method called SpikingLLM, which enables post-softmax quantization and ensures that QLLMs can be converted into fully-SNNs. To further enhance the performance of post-softmax quantization, we introduce QK2Head-migration module.

- We refine the ST-BIF[+] neuron to establish the equivalence between fully-SNNs and QLLMs. Then we introduce a window inhibition mechanism to address the over-firing issue of refined ST-BIF[+] neuron, which significantly improves the sparsity and reduces energy consumption.

- We convert SNN-unfriendly operators (*e.g.*, KV Cache, SiLU) to SNN-friendly versions counterparts, further enabling the equivalence between fully-SNNs and QLLMs.

- SpikingLLM achieves the state-of-the-art performance on perplexity and common-sense reasoning tasks with significant energy reduction (*e.g.*, compared to SpikeLLM(P) on Llama-2-7B in Table 2, our SpikingLLM improves the average accuracy of common-sense reasoning tasks by **26.37%** (47.79 ⇒ 60.28) with **60.34%** energy reduction (2.37J ⇒ 0.94J)).

## 2 RELATED WORKS

**Spiking Neural Networks.** The learning methods of SNNs come in twofolds: *directly training (DT)* and *ANN-to-SNN conversion (A2S)*. The DT algorithm leverages back-propagation through time (BPTT) (Wu et al., 2019) with surrogate gradient (Neftci et al., 2019) to update SNNs from scratch for a fixed time-step. However, the gap between SNNs and ANNs persists due to the gradient estimation error. Compared to DT algorithm, A2S algorithm leverages spiking neurons to replace the quantizers in quantized ANNs, leading to equivalent SNNs with comparable performance to ANNs (Wang et al., 2023; You et al., 2024b). Furthermore, A2S algorithm consumes less computational cost and time. However, most SNNs focus on computer vision tasks. As for language-oriented tasks, current SNNs (SpikeBERT (Lv et al., 2024), SpikingBERT (Bal & Sengupta, 2024), SpikeZIP-TF (You et al., 2024b), SpikeLM (Xing et al., 2024b) and SpikeGPT (Zhu et al., 2024b)) fail to scale up to the billion-level parameters. SpikeLLM (Xing et al., 2024a) scales up SNNs to billions of parameters, but their models are not fully-spiking. It remains a valuable issue to scale fully-spiking neural networks up to billions of parameters.

**Quantized Large Language Models.** Model quantization improves large language models (LLMs) efficiency by compressing weights and activations into lower bit-widths, reducing memory consumption and accelerating inference. *Quantization-aware training* (QAT), exemplified by LSQ (Esser et al., 2020) and U2NQ (Liu et al., 2022), achieves higher accuracy for smaller models through full retraining, and advances calibration-based techniques like EfficientQAT (Chen et al., 2024), further balancing efficiency and performance. *Post-training quantization* (PTQ) is more applicable on LLMs for its computational practicality, with methods like GPTQ (Frantar et al., 2023), SpQR (Dettmers et al., 2023), and AWQ (Lin et al., 2024a) focusing on weight compression, while SmoothQuant (Xiao et al., 2024), RPTQ (Yuan et al., 2023) and OmniQuant (Shao et al., 2024) jointly quantize weights and activations. However, previous PTQ methods (*e.g.*, OmniQuant) mostly focus on dynamic quantization with quantization scale dynamically determined by input, which is difficult to tackle with spiking-version input. Although PrefixQuant (Chen et al., 2025) integrates prefixed tokens into static quantization, enabling low-bit precision for LLMs with high accuracy and efficiency, the overlook on post-q and post-softmax quantization (as depicted in $2^{\text{nd}}$ column in Figure 3) makes it unable to convert matrix products of $QK^T$ and $\text{softmax}(\frac{QK^T}{\sqrt{d}})V$ into spiking matrix products. Consequently, it remains a challenge to establish specific QLLMs which are suitable to be converted into fully-SNNs.

## 3 Problem Formulation

In this section, we firstly introduce the paradigm of A2S algorithm. Then we bring in the current state-of-the-art QLLMs (PrefixQuant (Chen et al., 2025)) and illustrate its applicability to the A2S algorithm. Finally we propose the definition of full-spiking and clarify the intuition of SpikingLLM.

### 3.1 A2S Algorithm

A2S Algorithm transfers the parameters of the pre-trained ANNs into their SNNs counterpart while maintaining the synaptic connections in ANNs, which yields close-to-ANNs accuracy. In SpikingLLM, we inherit the A2S conversion algorithm from SpikeZIP-TF (You et al., 2024b) including the ANNs (LLMs) $\rightarrow$ QANNs (QLLMs) $\rightarrow$ SNNs conversion paradigm (as shown in Figure 2). For conversion paradigm, we insert activation quantizers in front of all the matrix products in ANNs (LLMs). SpikeZIP-TF leverages the quantization-aware training (QAT) method to achieve corresponding QANNs, which is com-

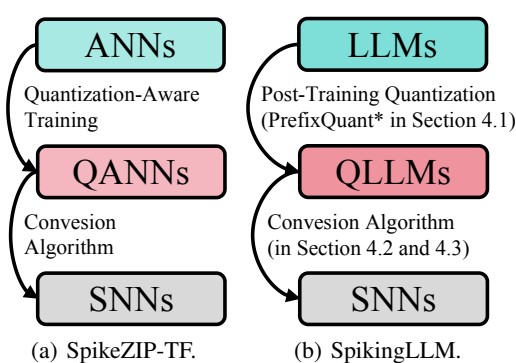

(a) SpikeZIP-TF.  (b) SpikingLLM.

Figure 2: SpikeZIP-TF and SpikingLLM.

putationally inefficient for LLMs. Consequently, we apply efficient post-training quantization (PTQ) method (PrefixQuant$^*$ in Section 4.1 ) to achieve corresponding QLLMs. Then we propose the conversion algorithm in Section 4.2 and Section 4.3 to replace the inserted quantizers with spiking neurons and ensure that all matrix products and operators can be converted to their spiking version.

### 3.2 PrefixQuant

PrefixQuant (Chen et al., 2025) introduces an efficient **static** quantization framework tailored to large language models, specifically focusing on **prefixed tokens** to enhance performance. By setting specific prefixed tokens in the KV cache, PrefixQuant eliminates token-wise outliers in linear inputs and Q/K/V, enhancing compatibility with per-tensor static quantization. When tackling spiking version input (which means we cannot acquire the total input at the current inference time-step), **static** quantization with fixed quantization parameters is more suitable to the A2S algorithm compared to **dynamic** quantization method (such as OmniQuant (Shao et al., 2024)) where the quantization parameter is dynamically determined by input. Consequently, we construct SpikingLLM on the basis of **static** quantization (PrefixQuant) rather than **dynamic** quantization (OmniQuant).

### 3.3 Fully-Spiking Definition

Inspired by the concept of spike-driven introduced by DT algorithm (Yao et al., 2023), we introduce the definition of **fully-spiking** for A2S algorithm, which means that **all operators in SNNs are in an event-driven or spiking version (calculation is triggered by spikes)**. However, current

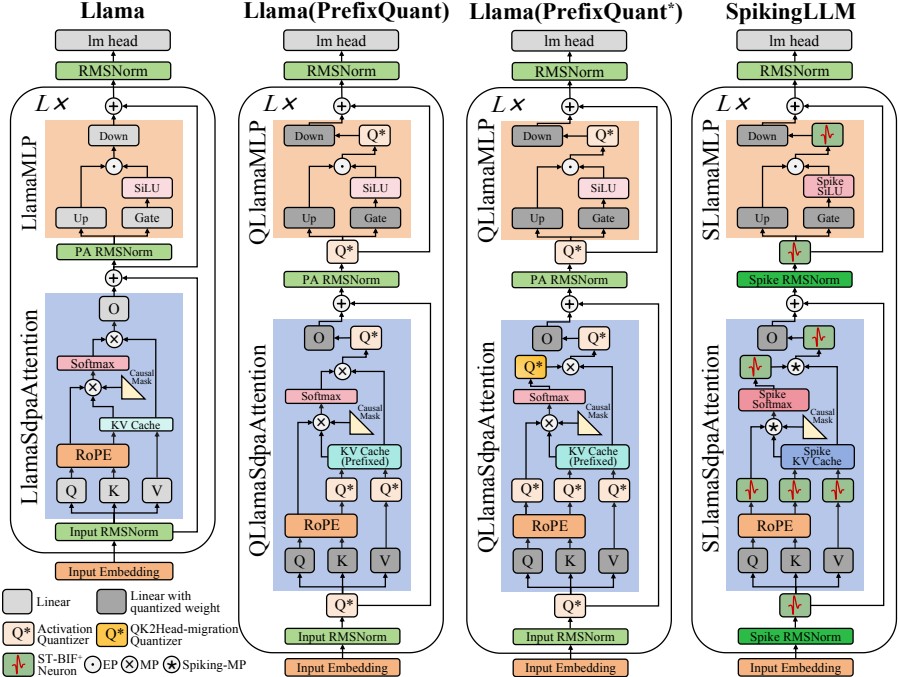

Figure 3: **Architecture of Llama, PrefixQuant, PrefixQuant\* and SpikingLLM.** PA, EP and MP refer to post-attention, element-wise product and matrix product, respectively. Compared to PrefixQuant, *PrefixQuant\* inserts post-q and post-softmax (QK2Head-migration) quantization to ensure that each matrix product can be converted to spiking matrix product.* SpikingLLM firstly substitutes ST-BIF+ neuron for all quantizers, then replaces SNN-unfriendly operators (Softmax, RMSNorm, SiLU and KV Cache (Prefixed)) with SNN-friendly ones.

QLLMs methods, such as OmniQuant (Shao et al., 2024) and PrefixQuant (Chen et al., 2025), overlook the quantization of query and softmax output (as depicted in 2nd column in Figure 3). As a result, the matrix products of $QK^T$ and $\text{softmax}(\frac{QK^T}{\sqrt{d}})V$ cannot be converted into spiking version. Additionally, operators in QLLMs (*e.g.*, KV Cache, SiLU) need to be converted into their spiking version. Although SpikeLLM (Xing et al., 2024a) introduces a spiking mechanism tailored to salient channels, operators on non-salient channels remain non-spiking version. Consequently, our SpikingLLM is the first to establish the equivalence between fully-SNNs and QLLMs.

## 4 METHODOLOGY

In this section, we firstly introduce post-q and post-softmax quantization on top of PrefixQuant to establish PrefixQuant\* (as shown in Figure 3) to ensure that all matrix products are equally converted into spiking matrix products. To further enhance the performance of PrefixQuant\*, we propose QK2Head-migration quantization, a novel approach that shifts the difficulties of post-softmax quantization from query and key dimension to head dimension. Then, we refine the ST-BIF+ neuron to make it fully equivalent to the quantizer in PrefixQuant\*. With the equivalence above, we introduce a window inhibition mechanism to further improve the sparsity of the refined ST-BIF+ neuron. Finally, we describe the design of SNN-friendly spike operators in SpikingLLM including Spike KV Cache.

### 4.1 PREFIXQUANT\* WITH QK2HEAD-MIGRATION QUANTIZATION

We firstly introduce PrefixQuant\* (3rd column in Figure 3) which inserts post-q and post-softmax quantization on the basis of PrefixQuant. For post-q quantization, we follow the post-k and post-v quantization in PrefixQuant. For 4-dimensional post-softmax output, we propose a novel strategy called *QK2Head-migration quantization*. As illustrated in Figure 4, QK2Head-migration quantization divides the softmax output into prefixed part and normal part. The prefixed part corresponds to the attention scores associated with the prefixed tokens introduced by PrefixQuant, while the normal part

Figure 4: **Architecture of QK2Head-migration Quantization.** Note that **T**, **R**, **A**, **C**, **Q** are the abbreviation of Transpose, Reshape, Accumulate, Calibration and Quantization. Stat calib input means the Pile (Gao et al., 2020) data distribution for static quantization parameter calibration. After **T**, **R** and **A** on stat calib input, quantized item stat is achieved to initialize quantization scale.

represents the standard attention scores computed during the forward pass, which depends on the input sequence.

To cope with the quantization of these two parts, we introduce query dimension migration ratio $q_m$ and key dimension migration ratio $k_m$ to redistribute the quantization complexity from query and key dimension to head dimension. Specifically, for both Prefixed and Normal parts, we firstly transpose, reshape and accumulate the stat calibration input to derive the quantized item stat. Then we initialize separate quantization scales for prefixed ($\text{Scale}_P$) and normal ($\text{Scale}_N$) parts, ensuring that each part is quantized optimally based on its stat calibration input. After calibration, we follow the same procedure to process the data input and achieve corresponding quantized data output. Then we leverage block-wise fine-tuning (Shao et al., 2024; Chen et al., 2024) to fine-tune PrefixQuant*.

## 4.2 ST-BIF+ NEURON REFINEMENT AND WINDOW INHIBITION MECHANISM

In SpikingLLM, we follow the ST-BIF+ neuron proposed in SpikeZIP-TF (You et al., 2024b), whose accumulated spikes (neuron output) is fully equivalent to the quantized activation. The quantization scale of the quantizer used in SpikeZIP-TF is a simple scalar, which is effective for quantizing models with limited parameters. However, when it comes to the quantization on large-scale LLMs, the quantization scale of quantizer Q* in PrefixQuant* is a matrix with group size. Consequently, we refine the ST-BIF+ neuron in SpikeZIP-TF to make it fully equivalent to Q*. Overview of the refined ST-BIF+ neuron is shown in Figure 5(a) and the equation of Q* is described in Equation (1). The notations are specified in Table 1.

Table 1: Summary of notations used in this paper.

| Notation | Description |
| --- | --- |
| $\vec{q}$ | quantization scale of quantizer Q* |
| $x$ | input of quantizer Q* |
| $V_t$ | membrane potential of neuron at $t$ time-step |
| $\vec{V}_{thr}$ | threshold voltage for neuron to fire a spike |
| $V^{in}, V^{out}$ | input or output voltage of neuron |
| $S_t$ | spike tracer at time-step $t$ |
| $S_{max}$ | maximum value in spike tracer |
| $\text{clip}(x, \alpha_{min}, \alpha_{max})$ | clip function that limits $x$ within $\alpha_{min}$ and $\alpha_{max}$ |
| $\Theta(V, V_{thr}, S)$ | output spike decision function of ST-BIF+ |
| $T_{eq}$ | time-step that SNNs enter the equilibrium state |
| $n$ | head number of softmax output |
| $\text{dim}_s, \text{dim}_d$ | dimension of stat calib input and data input |
| $q_m, k_m$ | migration ratio for query, key dimension |
| $\text{Scale}_P, \text{Scale}_N$ | quantization scale for Prefixed and Normal part |
| $\mathcal{C}[\cdot], \mathcal{Z}(\cdot)$ | concatenate and zeros like operator |
| $T_P, T_S$ | prefixed and stored tokens |
| $T_t$ | tokens at t time-step |
| $L$ | length of inhibiting window |

$$x \xrightarrow[\text{Group}]{\text{Reshape}} \hat{x}; \text{Quantize}(\hat{x}) = \vec{q} \cdot \text{clamp}(\text{round}(\hat{x}/\vec{q}), \alpha, \beta); \text{Quantize}(\hat{x}) \xrightarrow[\text{Reshape}]{\text{Regroup}} O_q \quad (1)$$

As for the refined ST-BIF+ neuron, the dynamics can be expressed as follows (note that the threshold voltage $\vec{V}_{thr}$ is equal to quantization scale $\vec{q}$):

$$V_t^{in} \xrightarrow[\text{Group}]{\text{Reshape}} \hat{V}_t^{in}; \Theta(V, \vec{V}_{thr}, S) = \begin{cases} 1; & V \geq \vec{V}_{thr} \ \& \ S < S_{max} \\ 0; & \text{other} \\ -1; & V < 0 \ \& \ S > S_{min} \end{cases};$$

$$V_t = V_{t-1} + \hat{V}_t^{in} - \vec{V}_{thr} \cdot \Theta(V_{t-1} + \hat{V}_t^{in}, \vec{V}_{thr}; S_{t-1}); S_t = S_{t-1} + \Theta(V_{t-1} + \hat{V}_t^{in}; \vec{V}_{thr}; S_{t-1})$$

$$S_{T_{eq}} \cdot \vec{V}_{thr} \xrightarrow[\text{Reshape}]{\text{Regroup}} O_s$$

(2)

The accumulation of spikes from the refined ST-BIF+ neuron $O_s$ is equivalent to the output of Q* $O_q$.

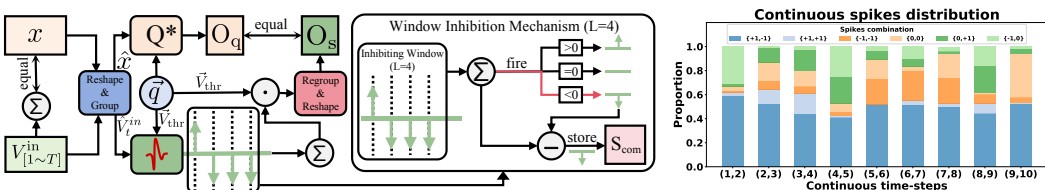

(a) Refined ST-BIF⁺ neuron with window inhibition mechanism.

(b) Continuous spikes distribution.

Figure 5: **Refined ST-BIF⁺ neuron with window inhibition mechanism and continuous spikes distribution.**

We plot the continuous spikes distribution under continuous time-steps in Figure 5(b) and find that $\{+1, -1\}$ makes up the majority of the continuous spikes combination. Specifically, most ST-BIF⁺ neurons tend to fire a positive (negative) spike to counteract the negative (positive) spike from the last time-step, leading to fire redundant spikes ("*over-firing*" issue). To address the over-firing issue and reduce the energy consumption, we propose window inhibition mechanism. As depicted in Figure 5(a), we introduce an inhibiting window (*e.g.*, window length L = 4) to accumulate the L time-step spikes to fire 1 time-step spike, which significantly suppresses the over-firing issue and improves the sparsity. For the equivalence between refined ST-BIF⁺ neuron with window inhibition mechanism and quantizer $Q^*$, we introduce a spike tracer $S_{com}$ to store the redundant spikes for compensation (*e.g.*, the sum of spikes in the inhibiting window is greater than 1 or less than -1). The detailed procedure of window inhibition mechanism is illustrated in appendix (Section A1).

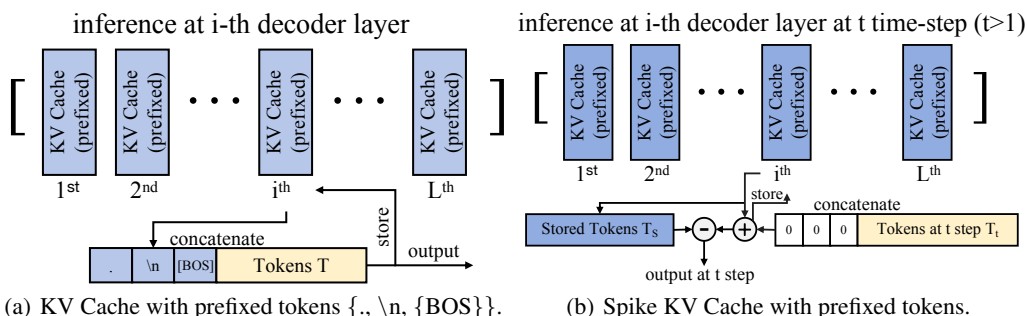

(a) KV Cache with prefixed tokens {., \n, {BOS}}.

(b) Spike KV Cache with prefixed tokens.

Figure 6: **Architecture of KV cache and Spike KV Cache with prefixed tokens.**

## 4.3 SNN-FRIENDLY SPIKE OPERATORS

To futher ensure the equivalence between PrefixQuant$^*$ and SpikingLLM, we introduce SNN-friendly spike operators (*e.g.*, Spike KV Cache, Spike SiLU.). As depicted in Figure 6(b), we introduce Spike KV Cache $S_{KV}$ with prefixed tokens to enable the KV Cache during SNNs inference. The inference of $S_{KV}$ at i-th decoder layer under t time-step is described in Equation (3). *At the first time-step* ($t = 1$), $S_{KV}$ outputs and stores the concatenated prefixed and original tokens, which is the same to the original KV Cache in Figure 6(a). *As for t time-step* ($t > 1$), $S_{KV}$ concatenates zeros tensors (with the same shape of prefixed tokens $T_P$) with tokens at t time-step $T_t$ to output, which ensures that the accumulation of $S_{KV}$ output equals to original KV Cache output. Then $S_{KV}$ stores the sum of output and stored tokens $T_S$ back to the cache.

$$
S_{KV}(T_t, T_P) = \begin{cases} \overbrace{\mathcal{C}[T_P, T_t]}^{\text{store}}; & t = 1 \\ \overbrace{\mathcal{C}[\mathcal{Z}(T_P), T_t] + T_S}^{\text{store}} - T_S; & t > 1 \end{cases} \tag{3}
$$

Since that Rotary Position Embedding (RoPE) operator is a linear mapping, the original RoPE is applicable to SpikingLLM. For Softmax, SiLU and RMSNorm, we follow the differential strategy from SpikeZIP-TF (You et al., 2024b) to introduce Spike Softmax, Spike SiLU and Spike RMSNorm. The detailed procedure of spike operators above are illustrated in appendix (Section A2).

Table 2: **Comparison on Llama-2 models.** T refers to inference time-step for SNNs. Best results are in **bold**, runner-up results are marked in `gray`. SpikeLLM(O) and SpikeLLM(P) refer to SpikeLLM with OmniQuant and PrefixQuant, respectively. WT2, HS and WG refer to Wikitext2, HellaSwag and Winogrande, respectively. Inhibiting window length L is set to 4 (L = 4).

| Method | Category | Fully-Spiking | T | Bits | Energy (J)(↓) | Perplexity(↓) | | Zero-shot Accuracy(↑) | | | | | |
| --- | --- | --- | --- | --- | --- | --- | --- | --- | --- | --- | --- | --- | --- |
| | | | | | | WT2 | C4 | PIQA | ARC-e | ARC-c | HS | WG | Avg. |
| **LLAMA-2-7B** | ANNs | – | – | FP16 | 20.02 | 5.47 | 6.97 | 79.11 | 74.62 | 46.25 | 76.00 | 69.22 | 69.04 |
| OmniQuant | QLLMs | – | – | W4A4 | 4.71 | 15.25 | 19.35 | 62.19 | 45.62 | 25.43 | 39.15 | 52.17 | 44.91 |
| PrefixQuant | | | | W4A4KV4 | 2.15 | 6.22 | – | 77.20 | 71.51 | 43.94 | 73.75 | 67.80 | 66.84 |
| PrefixQuant* | | | | W4A4QKVS4 | 1.64 | 11.56 | 14.10 | 72.85 | 62.50 | 36.95 | 68.01 | 61.96 | 60.45 |
| | | | | W4A5QKVS5 | 1.91 | 7.78 | 9.56 | 75.03 | 66.04 | 38.91 | 70.89 | 63.77 | 62.93 |
| SpikeLLM(O) | SNNs | ✗ | – | W4A4 | 5.18 | 11.46 | 14.45 | 62.79 | 51.01 | 27.13 | 43.47 | 53.83 | 47.65 |
| SpikeLLM(P) | | ✗ | – | W4A4QKV4 | 2.37 | 11.32 | 15.01 | 62.58 | 50.93 | 27.11 | 43.85 | 54.01 | 47.70 |
| SpikingLLM | | ✓ | 16 | W4A4QKVS4 | **0.94** | 10.99 | 13.78 | 73.45 | 61.95 | 36.43 | 68.02 | 61.56 | 60.28 |
| (L = 4) | | | 32 | W4A5QKVS5 | 1.53 | **7.71** | **9.35** | **74.54** | **65.66** | **39.16** | **71.22** | **62.51** | **62.26** |
| **LLAMA-2-13B** | ANNs | – | – | FP16 | 38.50 | 4.88 | 6.46 | 80.52 | 77.48 | 49.06 | 79.37 | 72.22 | 71.73 |
| OmniQuant | QLLMs | – | – | W4A4 | 9.16 | 12.40 | 15.87 | 67.03 | 53.96 | 30.55 | 62.91 | 44.83 | 51.86 |
| PrefixQuant | | | | W4A4KV4 | 4.18 | 6.22 | – | 78.51 | 75.80 | 46.67 | 76.54 | 72.06 | 69.92 |
| PrefixQuant* | | | | W4A4QKVS4 | 3.16 | 7.99 | 10.68 | 75.57 | 69.49 | 41.47 | 72.45 | 65.27 | 64.85 |
| | | | | W4A5QKVS5 | 3.67 | 6.28 | 8.11 | 77.48 | 73.32 | 42.92 | 74.45 | 69.93 | 67.02 |
| SpikeLLM(O) | SNNs | ✗ | – | W4A4 | 10.07 | 9.56 | 12.48 | 65.29 | 55.81 | 28.41 | 48.13 | 55.56 | 50.64 |
| SpikeLLM(P) | | ✗ | – | W4A4QKV4 | 4.39 | 9.94 | 12.59 | 65.93 | 55.89 | 28.73 | 48.22 | 55.21 | 50.80 |
| SpikingLLM | | ✓ | 16 | W4A4QKVS4 | **2.08** | 7.80 | 10.32 | 76.50 | 70.41 | 41.98 | 72.42 | 65.51 | 65.36 |
| (L = 4) | | | 32 | W4A5QKVS5 | 3.07 | **6.26** | **8.07** | **77.26** | **73.44** | **43.43** | **74.37** | **66.69** | **67.64** |

# 5 EXPERIMENTS

## 5.1 SETUPS

**Training Details.** We follow the fine-tuning setting from PrefixQuant (Chen et al., 2025) to fine-tune PrefixQuant*. During fine-tuning, we optimize the block-wise output mean square error. We use 512 samples from Pile (Gao et al., 2020) with a 1024 context length as fine-tuning dataset. For **W**eight quantization, we choose 4-bit (denoted as W4). For **A**ctivation, **Q**uery, **K**ey, **V**alue and **S**oftmax quantization, we conduct experiments on 4-bit and 5-bit (denoted as A4QKVS4 and A5QKVS5, respectively) quantization. The fine-tuning batch size and number of epochs are set to 4 and 20, respectively. For QK2Head-migration quantization, we set $q_m$ to 1 and $k_m$ to 16. For SNNs inference, we set time-step T to 16 and 32 for A4QKVS4 and A5QKVS5, respectively. Finally we set inhibiting window length L = 4 during inference.

**Evaluation Tasks.** We evaluate SpikingLLM on Llama-2-7B, Llama-2-13B (Touvron et al., 2023), Llama-3-8B (Grattafiori & et al., 2024) and Mistral-7B (Jiang et al., 2023). We follow the evaluation methods from PrefixQuant and SpikeLLM as the primary baselines. We also conduct experiments on SpikeLLM with PrefixQuant (denoted as SpikeLLM(P)) in Table 2 for a fair comparison between SpikingLLM and SpikeLLM. Specifically, we evaluate the perplexity (PPL) of language generation on Wikitext2 (Merity et al., 2016) and C4 (Raffel et al., 2023) benchmarks. For zero-shot common-sense reasoning tasks, we evaluate SpikingLLM on PIQA (Bisk et al., 2019), ARC-easy (Clark et al., 2018), ARC-challenge (Clark et al., 2018), HellaSwag (Clark et al., 2018) and Winogrande (Sakaguchi et al., 2019). We report *acc* for WinoGrande and *acc_norm* for remaining datasets, following Qserve (Lin et al., 2024b). We also compare SpikingLLM with SpikeGPT (Zhu et al., 2023) and other efficient LLMs (MatMul-free LLM (Zhu et al., 2024a) and ShiftAddLLM (You et al., 2024a)) in appendix (Section A5 and Section A6).

**Energy Consumption Metric.** We inherit the operation metric proposed in SpikingFormer (Zhou et al., 2023) to calculate the Multiply-ACcumulate operations (MACs) and ACcumulate-Only operations (ACs) of self-attention and linear operators. For LLMs and QLLMs, we calculate the number of MACs #MACs. For SNNs, we calculate the number of ACs #ACs and MACs #MACs. Then we sample the weights and activations from different methods to estimate the average energy consumption of a single MAC operation $E_{MAC}$ and AC operation $E_{AC}$ (The detailed energy estimation procedure is illustrated in appendix (Section A3)). Finally we follow the formula $E_{SNNs} = \#ACs \times E_{AC} + \#MACs \times E_{MAC}$ and $E_{LLMs/QLLMs} = \#MACs \times E_{MAC}$ from Spiking-Former (Zhou et al., 2023) to estimate the total energy consumption for SNNs, LLMs and QLLMs.

Table 3: **Performance of SpikingLLM on Llama-3-8B and Mistral-7B.**

| Method | Category | Fully-Spiking | T | Bits | Energy (J)(↓) | Perplexity(↓) | | Zero-shot Accuracy(↑) | | | | | |
|---|---|---|---|---|---|---|---|---|---|---|---|---|---|
| | | | | | | WT2 | C4 | PIQA | ARC-e | ARC-c | HS | WG | Avg. |
| **LLAMA-3-8B** | ANNs | – | – | FP16 | 22.24 | 6.14 | 8.88 | 80.79 | 77.69 | 53.33 | 79.16 | 72.53 | 72.70 |
| PrefixQuant* | QLLMs | – | – | W4A4QKVS4 | 1.83 | 10.85 | 15.82 | 75.24 | 68.01 | 42.06 | 71.27 | 61.56 | 63.63 |
| | | | | W4A5QKVS5 | 2.12 | 8.20 | 11.61 | 77.48 | 73.65 | 46.76 | 75.60 | 67.32 | 68.16 |
| SpikingLLM (L = 4) | SNNs | ✓ | 16 | W4A4QKVS4 | **1.04** | 10.34 | 15.38 | 75.51 | 68.12 | 41.93 | 71.72 | 61.23 | 63.70 |
| | | | 32 | W4A5QKVS5 | 1.77 | **8.14** | **11.28** | **77.82** | **73.43** | **46.91** | **75.24** | **67.66** | **68.21** |
| **Mistral-7B** | ANNs | – | – | FP16 | 19.46 | 5.49 | 8.41 | 82.46 | 82.62 | 58.87 | 82.94 | 74.11 | 76.20 |
| PrefixQuant* | QLLMs | – | – | W4A4QKVS4 | 1.60 | 7.74 | 10.68 | 78.24 | 76.64 | 51.28 | 76.97 | 62.67 | 69.16 |
| | | | | W4A5QKVS5 | 1.86 | 6.39 | 9.20 | 81.23 | 80.30 | 56.57 | 80.94 | 71.51 | 74.11 |
| SpikingLLM (L = 4) | SNNs | ✓ | 16 | W4A4QKVS4 | **0.92** | 7.32 | 10.12 | 78.05 | 76.82 | 51.43 | 77.21 | 62.45 | 69.19 |
| | | | 32 | W4A5QKVS5 | 1.46 | **6.28** | **8.98** | **81.44** | **80.21** | **56.84** | **81.03** | **71.99** | **74.30** |

## 5.2 RESULTS COMPARISON

**Results on Perplexity Tasks.** Table 2 shows the experimental results on Llama-2-7B and Llama-2-13B. For perplexity metric on Wikitext2 and C4 benchmarks, SpikingLLM achieves equivalent results with PrefixQuant* under the same setting. For 4-bit quantization on Llama-2-7B, SpikingLLM surpasses SpikeLLM(P) by 0.33 on Wikitext2 and 1.23 on C4. For 4-bit quantization on Llama-2-13B, SpikingLLM outperforms SpikeLLM(P) by 2.14 on Wikitext2 and 2.27 on C4. Furthermore, SpikingLLM achieves state-of-the-art performance on both Wikitext2 and C4 benchmarks with 5-bit quantization. We also extend our SpikingLLM on Llama-3-8B and Mistral-7B in Table 3, which further verifies the equivalence between SpikingLLM and PrefixQuant*.

**Results on common-sense Reasoning Tasks.** As tabulated in Table 2, SpikingLLM achieves promising results on common-sense reasoning tasks. For 4-bit quantization, SpikingLLM achieves an average zero-shot accuracy of 60.28 on Llama-2-7B and 65.36 on Llama-2-13B, surpassing SpikeLLM(P) by 12.58 and 14.56, respectively. Moreover, with 5-bit quantization, SpikingLLM achieves an average zero-shot accuracy of 62.26 on Llama-2-7B and 67.64 on Llama-2-13B, further closing the gap between ANNs and SNNs. Notably, SpikingLLM outperforms PrefixQuant* on complex reasoning tasks such as ARC-c and HellaSwag. Furthermore, consistent experimental results on Llama-3-8B and Mistral-7B in Table 3 demonstrate the generalisability of SpikingLLM.

**Results on Energy Consumption.** Based on the experimental results in Table 2 and Table 3, SpikingLLM demonstrates significant advantages in reducing energy by effectively converting Multiply-ACcumulate operations (MACs) into ACcumulate-Only operations (ACs) through its fully-spiking paradigm. For instance, on Llama-2-7B model under 4-bit quantization, SpikingLLM achieves a remarkable reduction ($1.64 \Rightarrow 0.94$) compared with PrefixQuant*. Note that SpikeLLM exhibits higher energy consumption than corresponding QLLMs (*e.g.*, $2.15 \Rightarrow 2.37$ for PrefixQuant). This discrepancy arises that SpikeLLM fail to embrace the fully-spiking paradigm, instead maintaining a hybrid approach which still relies on traditional Multiply-ACcumulate operations (MACs). We incorporate the detailed analysis on energy consumption in appendix (Section A3 and Section A4).

## 5.3 ABLATION STUDY

**Ablation on QK2Head-migration quantization.** As tabulated in Table 4, we test different settings of PrefixQuant*, including versions with/without QK2Head-migration quantization, to verify the effectiveness of QK2Head-migration quantization. Note that for PrefixQuant* without QK2Head-migration quantization, we reshape softmax output into 3-dimensional and quantize it through activation quantization in PrefixQuant.

Table 4: **Ablation on QK2Head-migration quantization and Spike KV Cache on Wikitext2.**

| QK2Head-migration | Spike KV Cache | W4A4QKVS4 | | W4A5QKVS5 | |
|---|---|---|---|---|---|
| | | 2-7B | 2-13B | 2-7B | 2-13B |
| ✗ | ✗ | 556.34 | 512.13 | 532.81 | 508.94 |
| ✓ | ✗ | 523.12 | 489.16 | 518.94 | 498.73 |
| ✗ | ✓ | 24.17 | 42.31 | 9.68 | 8.69 |
| ✓ | ✓ | **10.99** | **7.80** | **7.71** | **6.26** |

The results demonstrate that the introduction of QK2Head-migration quantization significantly enhances performance across all quantization scenarios for both Llama-2-7B and Llama-2-13B. For instance, for Llama-2-7B with W4A4QKVS4, Wikitext2 perplexity is reduced from 24.17 to 10.99 when QK2Head-migration quantization is applied. Similarly, for Llama-2-13B with W4A4QKVS4, Wikitext2 perplexity decreases from 42.31 to 7.80. These improvements highlight the substantial benefits of incorporating QK2Head-migration quantization.

**Ablation on Spike KV Cache.** We also present the experimental results of SpikingLLM with/without Spike KV Cache in Table 4. As detailed in Table 4, original KV Cache fails to process spiking inputs effectively, resulting in a fundamental discrepancy between SNNs and corresponding QLLMs. This limitation highlights the necessity of introducing Spike KV Cache, which ensures that the cumulative output aligns precisely with the output of KV Cache in QLLMs. For instance, in the cases of LLAMA-2-7B and LLAMA-2-13B with W4A4QKVS4, Wikitext2 perplexity are significantly reduced from 523.12 to 10.99 and 489.16 to 7.80 when Spike KV Cache is employed. These substantial reductions demonstrate the effectiveness of the Spike KV Cache in processing spiking input while preserving the consistency of output.

**Ablation on post-q and post-softmax quantization.** To verify the necessities of introducing post-q and post-softmax quantization, we compare the energy consumption of SpikingLLM with/without post-q and with/without post-softmax quantization on Llama-2 under W4A4QKVS4 in Table 5. As illustrated, the introduction of post-q and post-softmax quantization drastically reduces energy consumption. The significant reduction demonstrates that the introduction of post-q and post-softmax quantization enables the conversion of matrix products of $QK^T$ and $\mathrm{softmax}(\frac{QK^T}{\sqrt{d}})V$ into spiking matrix products, which effectively converts high-energy MACs into low-energy ACs.

Table 5: **Ablation on post-q and post-softmax quantization**.

| Post-q | Post-softmax | Fully-Spiking | Energy (J)($\downarrow$) | |
|---|---|---|---|---|
| | | | 2-7B | 2-13B |
| ✗ | ✗ | ✗ | 2.77 | 5.64 |
| ✓ | ✗ | ✗ | 2.12 | 4.02 |
| ✗ | ✓ | ✗ | 1.59 | 3.65 |
| ✓ | ✓ | ✓ | **0.94** | **2.08** |

**Ablation on Window Inhibition Mechanism.** We set multiple values to inhibiting window length L in Table 6 to verify the effectiveness of window inhibition mechanism. As illustrated, the energy consumption of SpikingLLM without window inhibition mechanism (L = 1) is 1.70J, which is comparable to the corresponding QLLMs (1.64J in Table 2). With the introduction of window inhibition mechanism, our SpikingLLM significantly improves sparsity (34.93% $\Rightarrow$ 63.21%) and reduces energy consumption (1.70J $\Rightarrow$ 0.94J) without performance degradation.

Table 6: **Ablation on window inhibition mechanism**.

| L | Sparsity($\uparrow$) | Energy (J)($\downarrow$) | LLAMA-2-7B WT2($\downarrow$) | C4($\downarrow$) |
|---|---|---|---|---|
| 1 | 34.93% | 1.70 | **10.91** | **13.68** |
| 2 | 49.78% | 1.19 | 10.94 | 13.71 |
| 4 | **63.21%** | **0.94** | 10.99 | 13.78 |

**Ablation on ($q_m$, $k_m$) Settings.** Figure 7 presents the heatmaps of Wikitext2 perplexity (PPL) results for Llama-2-7B model under various ($q_m$, $k_m$) settings on QK2Head-migration quantization. As depicted, the effectiveness of QK2Head-migration quantization varies significantly depending on the bit precision and ($q_m$, $k_m$) settings. For 4-bit quantization, the optimal performance (11.56) is achieved with ($q_m = 1$, $k_m = 16$) setting, which demonstrates the effectiveness of this setting in low-bit quantization. Similarly, for 5-bit quantization, ($q_m = 1$, $k_m = 16$) setting also delivers the best result, with a PPL of 7.78, further validating the robustness of this approach across different bit-widths.

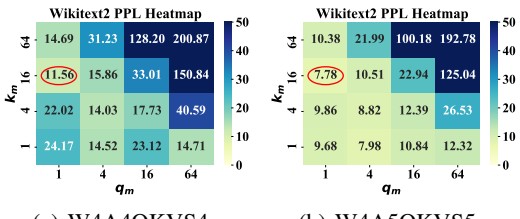

(a) W4A4QKVS4.     (b) W4A5QKVS5.

Figure 7: **Wikitext2 perplexity with Llama-2-7B under various ($q_m$, $k_m$) settings on QK2Head-migration quantization**.

## 6 CONCLUSION

SpikingLLM introduces an innovative ANN-to-SNN conversion method that establishes the equivalence between fully-spiking neural networks and quantized large language models. To make the equivalence applicable, we introduce QK2Head-migration quantization, refined ST-BIF+ with window inhibition mechanism and SNN-friendly spike operators. These advancements enable SpikingLLM to achieve state-of-the-art performance on both perplexity and common-sense reasoning tasks, while significantly reducing energy consumption. To the best of our knowledge, SpikingLLM is the first conversion-based method on fully-spiking large language models. We anticipate that SpikingLLM can be further extended to incorporate learning-based methods, which hold the potential to achieve even more promising performance while further reducing energy consumption.

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

# Appendix

## A1 WINDOW INHIBITION MECHANISM

---
**Algorithm 1** Window Inhibition Mechanism

---
**Input**: Input voltage t t time-step $\hat{V}_t^{\text{in}}$.
**Model**: Refined ST-BIF$^+$ Neuron $\Theta$.
**Parameter**: Spike Tracer at t time-step $S_t = 0$, Spike Tracer for previous window $S_{\text{pre}} = 0$, Spike Tracer for compensation $S_{\text{com}} = 0$, Inhibiting Window Length L, time-step T, Threshold voltage for neuron to fire a spike $\vec{V}_{\text{thr}}$, Membrane potential of neuron at t time-step $V_t$.
**Output**: Firing spikes spike.

1: **for** t = 1 to T **do**
2:    $S_t = S_{t\text{-}1} + \Theta(V_{t\text{-}1} + \hat{V}_t^{\text{in}}, \vec{V}_{\text{thr}}, S_{t\text{-}1})$
3:    # Window inhibition process
4:    **if** (t-1) % L == 0 **then**
5:      $\text{spike} = \begin{cases} 1; & S_t - S_{\text{pre}} > 0 \\ 0; & \text{other} \\ -1; & S_t - S_{\text{pre}} < 0 \end{cases}$
6:      # Update spikes for compensation
7:      $S_{\text{com}} = S_t - S_{\text{pre}} - \text{spike}$
8:    **else**
9:      # Block firing spikes
10:      $\text{spike} = \hat{V}_t^{\text{in}} * 0$.
11:      **if** (t-1) % L == 1 **then**
12:        # Update spike tracer for previous window
13:        $S_{\text{pre}} = S_t.\text{clone}()$
14:      **end if**
15:    **end if**
16: **end for**
17: # Firing compensated spikes
18: **while** $\max(S_{\text{com}}.\text{abs}())! = 0$ **do**
19:    $\text{spike} = \begin{cases} 1; & S_{\text{com}} > 0 \\ 0; & \text{other} \\ -1; & S_{\text{com}} < 0 \end{cases}$
     $S_{\text{com}} = S_{\text{com}} - \text{spike}$
20: **end while**

---

The detailed process of the window inhibition mechanism is specified in Algorithm 1. As illustrated, we introduce an inhibiting window with length L to combine the original L time-step spikes into 1 time-step spike (fire one positive spike if $S_t - S_{\text{pre}} > 0$, fire one negative spike if $S_t - S_{\text{pre}} < 0$). For the equivalence between refined ST-BIF$^+$ neuron with window inhibition mechanism and quantizer $Q^*$, we introduce a spike tracer $S_{\text{com}}$ to trace the redundant spikes for compensation (*e.g.*, the sum of spikes in the inhibiting window is greater than 1 or less than -1). When it comes to time-step t satisfying $(t - 1)\%L == 0$, the refined ST-BIF$^+$ neuron fires a spike. After firing the spike, the redundant spikes $S_t - S_{\text{pre}} - \text{spike}$ should be updated into $S_{\text{com}}$. After T time-step, $S_{\text{com}}$ should fire redundant spikes until there is no spike left. The introduction of window inhibition mechanism significantly suppresses the over-firing issue and improves the sparsity, leading to apparent reduction on energy consumption. Note that under most circumstances, there are no redundant spikes in $S_{\text{com}}$, which suggests that redundant spikes from inhibiting window can also counteract each other as time-step increases, further verifying the effectiveness of window inhibition mechanism.

## A2 SPIKE OPERATORS

Regarding the differential strategy (Figure A1) to convert the SNN-unfriendly operators (*e.g.*, Softmax, RMSNorm, SiLU) to SNN-friendly counterparts, the definition is as follows:

$$X_t = X_{t-1} + x_t; \quad O_t = \sigma(X_t)$$
$$O_{\mathcal{S},t} = O_t - O_{t-1} \tag{A1}$$

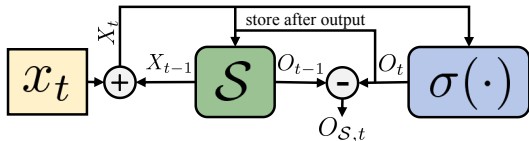

where $\sigma(\cdot)$ represents the ANN operators including Softmax, RMSNorm and SiLU. $x_t$ and $O_{\mathcal{S},t}$ are the input and output of the operator at timestep $t$ respectively, $X_t$ is the summation of the input during $t$ time-steps, $O_t$ is the output of the function $\sigma(\cdot)$ with input $X_t$. Both $X_t$ and $O_t$

Figure A1: **Architecture of spike operators (*e.g.*, Spike Softmax, Spike RMSNorm, Spike SiLU).** $\mathcal{S}$ refers to spike tracer. $\sigma(\cdot)$ refers to ANN operators (*e.g.*, Softmax, RMSNorm, SiLU).

are stored back to spike tracer $\mathcal{S}$ for computation at next time-step. The operators in ANNs can be made equivalent to its SNNs version by summing up $O_{\mathcal{S},t}$ through time.

## A3   ENERGY ESTIMATION PROCEDURE

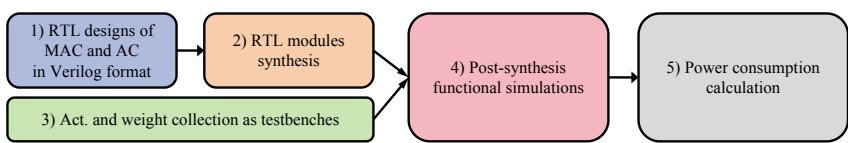

Figure A2: **Detailed procedure of energy estimation.**

We follow the standard EDA design flow (Kommuru & Mahmoodi, 2009) to evaluate energy consumption. The detailed evaluation procedure is illustrated in Figure A2 and summarized as follows:

1) We first implement the RTL designs of the MAC and AC units in Verilog format, following standard digital circuit design practices.

2) These RTL designs are synthesized into gate-level netlists using Synopsys Design Compiler, utilizing the TSMC 28nm HPC standard cell library.

3) We collect real activation and weight values from actual network inference and construct representative testbenches using these samples as input stimuli.

4) We perform post-synthesis functional simulations using Synopsys VCS, applying the testbenches to the synthesized netlists. The simulation generates VCD files that capture signal transitions and circuit switching activity over time.

5) We import the VCD files into Synopsys PrimeTime PX (Galbi et al., 2010), a gate-level power analysis tool, to calculate the dynamic power consumption based on real activity patterns and cell-level power models.

This procedure ensures that the reported energy values in this work are realistic and reflect actual data-dependent switching activity under typical network inference workloads. With the evaluation procedure above, we present the average energy consumption of a single MAC operation $E_{MAC}$ and a single AC operation $E_{AC}$ in Table A1. Note that the energy of 16-bit×16-bit Float-Point $E_{MAC}$ is adopted from (Tolliver et al., 2022). As a result, con-

Table A1: $E_{MAC}$ **and** $E_{AC}$ **estimation**.

| Operation | Energy (pJ) |
|---|---|
| 4-bit+4-bit Fixed-Point $E_{AC}$ | 0.0236 |
| 4-bit×4-bit Fixed-Point $E_{MAC}$ | 0.1141 |
| 4-bit×5-bit Fixed-Point $E_{MAC}$ | 0.1325 |
| 16-bit×16-bit Float-Point $E_{MAC}$ | 1.3900 |

verting MAC operations to AC operations with our fully-spiking neural networks can remarkably reduce over $80\%$ energy consumption.

We follow the procedure from Spikingformer (Zhou et al., 2023) to conduct energy evaluation between QLLMs and SNNs, which is concluded as follows:

$$E_{SNNs} = \#ACs \times E_{AC} + \#MACs \times E_{MAC}$$
$$E_{LLMs/QLLMs} = \#MACs \times E_{MAC} \tag{A2}$$

$\#MACs$ and $\#ACs$ refer to the total number of Multiply-ACcumulate and ACcumulate-Only operations, respectively. We follow the procedure from SpikeZIP-TF (You et al., 2024b) to calculate $\#MACs$ and $\#ACs$, which is concluded as follows:

---

**Algorithm 2** #MACs in Linear and Attention Layers

---

**Input**: Number of tokens N; Input feature dimension of linear layer $D_{in}$; Output feature dimension of linear layer $D_{out}$; Token dimension in attention layer D; Total layers L.
**Output**: #MACs

1: #MACs $\leftarrow 0$
2: **for** l = 1 to L **do**
3:   # Calculate #MACs for linear layer
4:   #MACs $\leftarrow$ #MACs $+ N \times D_{in} \times D_{out}$
5:   # Calculate #MACs for attention layer
6:   #MACs $\leftarrow$ #MACs $+ 2 \times N \times N \times D$
7: **end for**
8: **return** #MACs

---

**Algorithm 3** #ACs in Linear and Attention Layers

---

**Input**: Number of tokens N; Input feature dimension of linear layer $D_{in}$; Output feature dimension of linear layer $D_{out}$; Token dimension in attention layer D; Spike count per time-step $C_t$; Total inference time-steps T; Total layers L.
**Output**: #ACs

1: #ACs $\leftarrow 0$
2: **for** l = 1 to L **do**
3:   **for** t = 1 to T **do**
4:     **for** k = 1 to $C_t$ **do**
5:       # Linear layer: Dout synapses activated per spike and 2
         operations per spike for ST-BIF$^+$ neuron
6:       #ACs $\leftarrow$ #ACs $+ D_{out} + 2$
7:       # Attention layer: 2*N synapses activated per spike (dual
         matrix product) and 2 operations per spike for ST-BIF$^+$
         neuron
8:       #ACs $\leftarrow$ #ACs $+ 2 \times N + 2$
9:     **end for**
10:    **end for**
11: **end for**
12: **return** #ACs

---

Table A2: Detailed energy consumption estimation results on Llama-2-7B.

| Method | Category | Bits | MACs $(\times 10^{12})(\downarrow)$ | ACs $(\times 10^{12})(\downarrow)$ | Energy (J)$(\downarrow)$ | WT2 PPL$(\downarrow)$ |
|---|---|---|---|---|---|---|
| Llama-2-7B | ANNs | FP16 | 14.40 | 0. | 20.02 | 5.47 |
| PrefixQuant$^*$ | QLLMs | W4A4QKVS4 | 14.40 | 0. | 1.64 | 11.56 |
| | | W4A5QKVS5 | 14.40 | 0. | 1.91 | 7.78 |
| SpikeLLM(P) | | W4A4QKV4 | 13.45 | **11.44** | 2.37 | 11.32 |
| SpikingLLM(L=4) | SNNs | W4A4QKVS4 | **1.98** | 30.25 | **0.94** | 10.99 |
| | | W4A5QKVS5 | 2.14 | 52.82 | 1.53 | **7.71** |

The detailed energy consumption estimation results on Llama-2-7B are summarized in Table A2. Note that we reproduce PrefixQuant on SpikeLLM (denoted as SpikeLLM (P)), which neglects post-softmax quantization so that the energy consumption is a bit higher than PrefixQuant$^*$ with post-softmax quantization. For SpikeLLM (P), note that SpikeLLM neglects post-softmax quantization so that the softmax output remains 16-bit, #MACs consists of $13.01 \times 10^{12}$ 4-bit $\times$ 4-bit operation (0.0236pJ in Table A1) and $0.44 \times 10^{12}$ 16-bit $\times$ 16-bit operation (0.1141pJ in Table A1), the energy estimation is calculated as follows:

$$E_{\text{SpikeLLM (P)}} = 11.41 \times 0.0236 + 13.01 \times 0.1141 + 0.44 \times 1.39 = 2.37 \text{ J} \quad (A3)$$

For SpikingLLM, the energy estimation is calculated as follows:

$$E_{\text{SpikingLLM}} = 30.25 \times 0.0236 + 1.98 \times 0.1141 = 0.94 \text{ J} \quad (A4)$$

## A4   Analysis of energy consumption

We further clarify how SpikingLLM overcomes the specific disadvantages of SpikeLLM.

① SpikeLLM (Xing et al., 2024a) is based on dynamic quantization method OmniQuant (Shao et al., 2024), which is SNN-unfriendly due to the float calculation to determine quantization scale for each input during SNNs inference. Consequently, SpikeLLM fails to convert *Activation-Weight* (*aka.* **AW**) matrix product in the linear layer and *Activation-Activation* (*aka.* **AA**) matrix product in the attention layer into fully-spiking matrix product. However, the quantization scale of SpikingLLM is detemined during SNNs inference so that SpikingLLM effectively converts **AW** matrix product and **AA** matrix product into the accumulation of spikes as follows:

$$\text{for } \textbf{AW} \text{ matrix product: } O_{T_{eq}} = \vec{V}_{thr} \cdot \sum_{t=1}^{T_{eq}} W \cdot \Theta\left(x_t\right); \Theta\left(x_t\right) \in \{0, \pm 1\} \tag{A5}$$

$$\text{for } \textbf{AA} \text{ matrix product: } O_{T_{eq}} = \vec{V}_{thr}^{Q} \vec{V}_{thr}^{K} \sum_{t_1=1}^{T_{eq}} Q_{t_1} \cdot \sum_{t_2=1}^{T_{eq}} K_{t_2}$$

$$= \vec{V}_{thr}^{Q} \vec{V}_{thr}^{K} \sum_{t=1}^{T_{eq}} \Theta_Q\left(Q_t\right) \cdot K_t^{T} + Q_t \cdot \Theta_K^{T}\left(K_t\right) - \Theta_Q\left(Q_t\right) \cdot \Theta_K^{T}\left(K_t\right) \tag{A6}$$

$$\Theta_Q\left(Q_t\right), \Theta_K\left(K_t\right) \in \{0, \pm 1\}$$

Note that $W$ refers to weight, $x$ refers to input, $\Theta$ refers to refined ST-BIF$^+$ neuron, $\vec{V}_{thr}$ refers to threshold voltage in $\Theta$ (which is equal to quantization scale) and $T_{eq}$ refers to total time-step.

② SpikeLLM neglects post-softmax quantization so that SpikeLLM fails to convert matrix products between softmax $\left(\frac{QK^T}{\sqrt{d}}\right)$ and V into spiking matrix products. We propose QK2Head-migration post-softmax quantization to convert QLLMs to Fully-Spiking LLMs. The effectiveness of QK2Head-migration post-softmax quantization is verified in Table 4, Table 5 and Table A2, respectively.

## A5   Comparison between SpikingLLM and SpikeGPT

Table A3: **Comparison between SpikingLLM and SpikeGPT**. DT, PS, Time and WT2 PPL refer to directly training, parameter size, training time and wikitext2 perplexity respectively.

| Method | Category | Model | PS(B) | Bits | T($\downarrow$) | WT2 PPL($\downarrow$) | Energy (mJ)($\downarrow$) | GPU | Time($\downarrow$) |
|---|---|---|---|---|---|---|---|---|---|
| SpikeGPT | DT | SpikeGPT with Pre-training | 0.2 | – | 50 | 18.01 | 47.82 | 4 NVIDIA-V100 | 48 hours |
| SpikingLLM (L=4) | A2S | MobileLLM | 0.3 | W4A5QKVS5 | 32 | 14.56 | **22.36** | 1 NVIDIA-4090 | **52 seconds** |
| | | | | W8A5QKVS5 | 32 | 13.21 | 34.61 | | 61 seconds |
| | | Llama-3.2 | 1.0 | W4A4QKVS4 | **16** | 12.03 | 131.82 | | 72 seconds |
| | | | | W4A5QKVS5 | 32 | **10.97** | 223.87 | | 78 seconds |

We conduct experiments between SpikingLLM and SpikeGPT on WikiText2 perplexity in Table A3. For fair comparison, we choose MobileLLM-350M (Liu et al., 2024) with comparable parameter of SpikeGPT 216M With Pre-training as our ANN model. As tabulated, our SpikingLLM achieves lower Wikitext2 perplexity with lower time-step and energy consumption under the configuration of 5-bit quantization on Activation, Query, Key, Value and Softmax. Our SpikingLLM can also scale up to Large Language Models with billions parameters (*e.g.*, Llama-2-7B, Llama-2-13B and Llama-3.2-1B). We also compare the computational cost between SpikingLLM and SpikeGPT, compared with directly training method SpikeGPT, our SpikingLLM significantly reduces the computational cost.

## A6   Comparison between SpikingLLM and Other Efficient LLMs

We conduct experiments between SpikingLLM and other efficient LLMs, such as MatMul-free LLM (Zhu et al., 2024a) and ShiftAddLLM (You et al., 2024a). We first conduct experiments

Table A4: **Comparison between SpikingLLM and MatMul-free LLM**. M-LLM refers to MatMul-free-LLM. HS and WG refer to HellaSwag and Winogrande, respectively.

| Method | Model | PS(B) | MatMul-free | Bits | PIQA | ARC-e | ARC-c | HS | WG | Avg.(↑) | GPU | Time(↓) |
|---|---|---|---|---|---|---|---|---|---|---|---|---|
| MatMul-free LLM | M-LLM-370M | 0.3 | ✓ | – | 63.0 | 42.6 | 23.8 | 32.8 | 49.2 | 42.3 | 8 NVIDIA-H100 | 5 hours |
| SpikingLLM(ours) | MobileLLM | 0.3 | ✓ | W4A5QKVS5 | 63.3 | 42.4 | 24.7 | 43.8 | 53.3 | 45.5 | 1 NVIDIA-4090 | **52 seconds** |
| | | | | W8A5QKVS5 | **64.8** | **43.9** | **25.9** | **45.1** | **53.5** | **46.6** | | 61 seconds |
| MatMul-free LLM | M-LLM-1.3B | 1.3 | ✓ | – | 68.4 | 54.0 | 25.9 | 44.9 | 52.4 | 49.1 | 8 NVIDIA-H100 | 84 hours |
| | M-LLM-2.7B | 2.7 | ✓ | | 71.1 | 58.5 | 29.7 | 52.3 | 52.1 | 52.7 | | 173 hours |
| SpikingLLM(ours) | Llama-3.2-1B | 1 | ✓ | W4A4QKVS4 | 68.6 | 58.2 | 29.9 | 55.6 | 53.8 | 53.2 | 1 NVIDIA-4090 | **72 seconds** |
| | Llama-2-7B | 7 | ✓ | W4A4QKVS4 | **73.5** | **62.0** | **36.4** | **68.0** | **61.6** | **60.3** | | 269 seconds |

Table A6: **Comparison with the Llama-2-70B model of SpikeLLM**. T refers to inference time-step for SNNs. Best results are in **bold**, runner-up results are marked in gray .

| Method | Category | Fully-Spiking | T | Bits | Perplexity(↓) | | Zero-shot Accuracy(↑) | | | | | |
|---|---|---|---|---|---|---|---|---|---|---|---|---|
| | | | | | WT2 | C4 | PIQA | ARC-e | ARC-c | HSg | WG | Avg. |
| *LLAMA-2-70B* | | | | | | | | | | | | |
| SpikeLLM | SNNs | ✗ | – | W2A16 | 6.35 | 9.62 | 76.44 | 66.92 | 38.31 | 51.86 | 59.19 | 58.54 |
| *LLAMA-2-13B* | | | | | | | | | | | | |
| SpikingLLM | SNNs | ✓ | 32 | W4A5QKVS5 | **6.26** | **8.07** | 77.26 | 73.44 | 43.43 | 74.37 | 66.69 | 67.64 |
| *Mistral-7B* | | | | | | | | | | | | |
| SpikingLLM | SNNs | ✓ | 32 | W4A5QKVS5 | 6.28 | 8.98 | **81.44** | **80.21** | **56.84** | **81.03** | **71.99** | **74.30** |

between SpikingLLM and MatMul-free LLM in Table A4, our SpikingLLM surpasses MatMul-free LLM on zero-shot common-sense reasoning tasks with both millions and billions parameters models. Matmul-free LLM (Zhu et al., 2024a) leverages ternary weights to eliminate matrix multiplication in dense layers while optimizing the Gated Recurrent Unit (GRU) (Cho et al., 2014) to remove matrix multiplication from self-attention. The idea that leveraging ternary weights to eliminate matrix multiplication is similar to our refined ternary value(-1, 0, +1) ST-BIF$^+$ neuron, but our refined ST-BIF$^+$ neuron is introduced to replace activation quantizer. The effectiveness of our SpikingLLM on Matmul-free LLM is that SpikingLLM eliminates matrix multiplication through replacing activation quantizers in quantized large language models with equivalent ST-BIF$^+$ neurons, so that SpikingLLM don't need additional training like Matmul-free LLM. We also compare the computational cost between Matmul-free LLM and SpikingLLM in Table A4, our SpikingLLM significantly reduces the computational cost.

We then compare our SpikingLLM with ShiftAddLLM on WikiText2 perplexity in Table A5, our SpikingLLM surpasses ShiftAddLLM on LLMs with both millions and billions parameters. Note that ShiftAddLLM (You et al., 2024a) introduces shift-and-add operations to eliminate weight-activation multiplications, the key lim-

Table A5: **Comparison between ShiftAddLLM and SpikingLLM on WikiText2 Perplexity**.

| Method | Model | PS(B) | MatMul-free | Bits | WT2 PPL(↓) |
|---|---|---|---|---|---|
| ShiftAddLLM | OPT (Zhang et al., 2022) | 0.3 | ✗ | W2A16QKVS16 | 40.24 |
| SpikingLLM(ours) | MobileLLM | 0.3 | ✓ | W4A5QKVS5 | 14.56 |
| | | | | W8A5QKVS5 | **13.21** |
| ShiftAddLLM | Llama-2-7B | 7 | ✗ | W2A16QKVS16 | 8.11 |
| SpikingLLM(ours) | | | ✓ | W4A5QKVS5 | **7.71** |
| ShiftAddLLM | Llama-2-13B | 13 | ✗ | W2A16QKVS16 | 6.77 |
| SpikingLLM(ours) | | | ✓ | W4A5QKVS5 | **6.26** |

itation is its inability to eliminate activation-activation multiplications (e.g., $QK^T$ in self-attention layers). Compared to ShiftAddLLM, our SpikingLLM eliminates both weight-activation and activation-activation matrix multiplications through replacing activation quantizers in quantized large language models with equivalent refined ST-BIF$^+$ neurons, constructing matmul-free fully-spiking large language models.

## A7    COMPARISON WITH THE LLAMA-2-70B MODEL OF SPIKELLM

To further demonstrate the effectiveness of our SpikingLLM method, we compared the Llama-2-13B and Mistral-7B models of SpikingLLM with the Llama-2-70B model of SpikeLLM. Table A6 indicates that, even with fewer parameters, our SpikingLLM surpasses SpikeLLM on all perplexity and common-sense reasoning tasks, which further verifies the effectiveness of SpikingLLM.

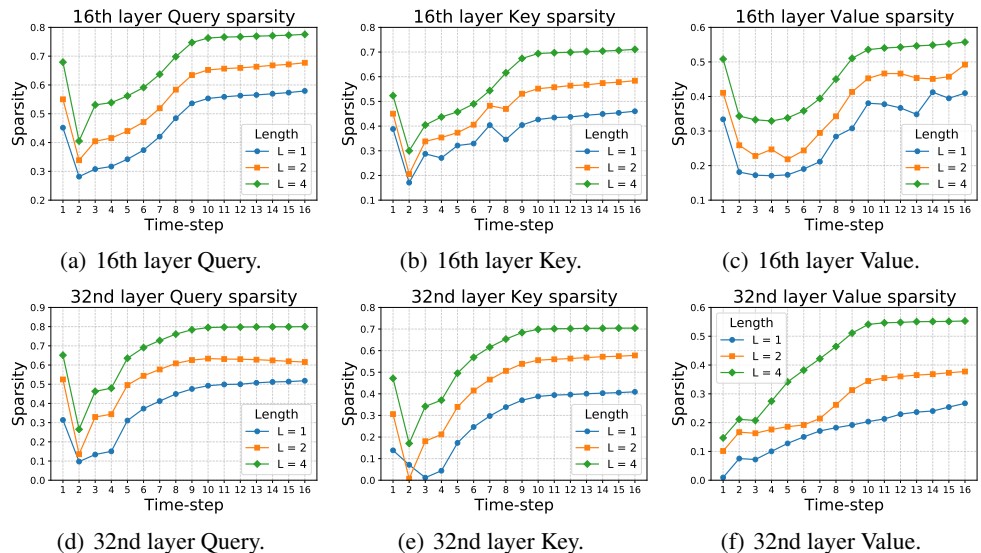

(a) 16th layer Query.     (b) 16th layer Key.     (c) 16th layer Value.

(d) 32nd layer Query.     (e) 32nd layer Key.     (f) 32nd layer Value.

Figure A3: **Sparsity of 16th and 32nd layer Query, Key, Value in Llama-2-7B with each time-step under different inhibiting window lengths.** All visualizations are sampled under W4A4QKVS4.

## A8  Sparsity Visualization

As depicted in Figure A3, we visualize the sparsity of 16th and 32nd layer Query, Key, Value in Llama-2-7B, which intuitively demonstrates the effectiveness of window inhibition mechanism. Note that, the improvement in sparsity and the reduction in energy consumption are more significant as inhibiting window length L increases.

## A9  Comparison between SpikingLLM and SpikeZIP-TF

Table A7: Comparison between LSQ and PrefixQuant[*] on Llama-2-7B.

| Model | Quantization Method | Quantization Type | GPU | Time | Bits | WT2 PPL |
|---|---|---|---|---|---|---|
| Llama-2-7B | LSQ | QAT | 4*NVIDIA-4090 | 6 hours | W4A4QKVS4 | 45.28 |
| | PrefixQuant[*] | PTQ | **1*NVIDIA-4090** | **269 seconds** | | **11.56** |

Apart from adapting the A2S conversion method in SpikeZIP-TF (You et al., 2024b) to PrefixQuant framework, we propose three innovations:

① To effectively achieve promising QLLMs, we insert post-q quantization and propose QK2Head-migration post-softmax quantization(in Section 4.1) to establish PrefixQuant[*] (As shown in Figure 3). As illustrated in Table A7, compared to Quantization-Aware Training (QAT) method LSQ (Esser et al., 2020) in SpikeZIP-TF, our PrefixQuant[*] effectively achieves QLLMs with promising performance.

② To establish the equivalence between QLLMs and SNNs, we firstly refine the ST-BIF[+] neuron in Section 4.2 to make it fully equivalent to quantizer in PrefixQuant[*] (quantizer with group-size matrix quantization scale). Then we propose SNN-friendly operators in SpikingLLM including Spike KV Cache (in Section 4.3), Spike Softmax, Spike SiLU and Spike RMSNorm (in Section A2).

③ In order to suppress redundant continuous $\{\pm1\}$ spikes from ST-BIF[+] neuron, we propose window inhibition mechanism in Section 4.2, which significantly improves the sparsity without performance degradation. As illustrated in Table 6 and Table A8, the introduction of window inhibition mechanism significantly improves sparsity and reduces energy consumption without performance degradation.

Table A8: **Ablation on window inhibition mechanism**.

| L | Sparsity(↑) | Energy (J)(↓) | LLAMA-2-13B WT2(↓) | C4(↓) |
|---|---|---|---|---|
| 1 | 32.82% | 3.96 | **7.72** | **10.23** |
| 2 | 48.94% | 2.66 | 7.75 | 10.26 |
| 4 | **62.51%** | **2.08** | 7.80 | 10.32 |

To conclude, our SpikingLLM advances SpikeZIP-TF by tailoring the conversion process to LLM-specific challenges (*e.g.*, effective PTQ on LLMs with post-softmax quantization, SNN-friendly LLMs operators, refined ST-BIF⁺ neuron with window inhibition mechanism to reduce energy consumption) and achieving the first fully-spiking billion-parameter language models.

## A10   ANALYSIS OF OUTLIER TOKENS ON SPIKINGLLM AND PREFIXQUANT.

We further analyze the outlier tokens between SpikingLLM and PrefixQuant in Table A9. We follow the definition of outlier tokens in PrefixQuant (Chen et al., 2025) to detect outlier tokens. Given token-wise maximum values $\mathbf{M} \in \mathbb{R}^T$, which represents the maximum values of each token. Then, outlier token in the i-th index of token sequence is identified when the ratio of their maximum values to the median of all maximum values exceeds a threshold $\eta$:

Table A9: **Comparison on outlier tokens between SpikingLLM and PrefixQuant**.

| Model | Method | Prefixed token | |
|---|---|---|---|
| | | Number | Content |
| Llama-2-7B | PrefixQuant | 3 | . \n [BOS] |
| | SpikingLLM | 3 | . \n [BOS] |
| Llama-2-13B | PrefixQuant | 3 | . the [BOS] |
| | SpikingLLM | 3 | . the [BOS] |
| Llama-3-8B | PrefixQuant | 1 | [BOS] |
| | SpikingLLM | 1 | [BOS] |
| Mistral-7B | PrefixQuant | 4 | . \n to [BOS] |
| | SpikingLLM | 4 | . \n to [BOS] |

$$\frac{\mathbf{M}_i}{\text{median}(\mathbf{M})} > \eta \qquad (A7)$$

where $\mathbf{M}_i$ is the maximum value of the i-th token, $\text{median}()$ denotes the function to find the median value from the vector. We then leverage the same calibration dataset Pile (Gao et al., 2020) and set the same outlier threshold $\eta = 64$ to determine outlier tokens before Post-Training Quantization (PTQ). Consequently, as shown in Table A9, the introduction of post-q and QK2Head-migration post-softmax quantization does not change the outlier tokens for the same model.

## A11   USE OF LLMS

We leverage LLMs to aid or polish writing. Specifically, LLMs help us find some grammar and spelling mistakes after we finish writing.

