# OpenReview forum: "SpikingLLM: A Conversion-Based Method with Window Inhibition Mechanism for Spiking Large Language Models"
_ICLR.cc/2026/Conference — Submitted to ICLR 2026_

### Official Review · Reviewer_4kKu · 2025-10-31

**Soundness:** 2
**Presentation:** 3
**Contribution:** 2
**Rating:** 4
**Confidence:** 3

**Summary:**

This paper introduces SpikingLLM, a conversion-based framework that transforms quantized large language models into fully spiking neural networks. The goal is to drastically reduce the energy cost of large models while keeping their performance competitive. The authors propose several key ideas to make this possible: a QK2Head-migration quantization technique to handle post-softmax quantization, a refined ST-BIF+ neuron equipped with a window inhibition mechanism to prevent over-firing, and SNN-friendly versions of traditionally non-spiking components such as the KV Cache. Experimental results on LLaMA and Mistral models show that SpikingLLM can cut energy use by over 60% while maintaining strong accuracy on language modeling and reasoning benchmarks.

**Strengths:**

1. The author provides solid empirical results across multiple large models.

2. The proposed technical solutions, like QK2Head-migration and window inhibition that seem effective.

**Weaknesses:**

1. The window inhibition mechanism suppresses firing within a short temporal window to prevent over-firing, but this changes the temporal distribution of spikes. Since the quantizer and neuron are tightly coupled, any mismatch between inhibition strength and quantization scale could distort activation magnitude.

2. Replacing continuous operators like softmax, SiLU, and KV Cache with “spiking-friendly” versions is a major modification, but the paper doesn’t quantify how much numerical or representational error each conversion introduces. The authors use “differential strategies” from SpikeZIP-TF, but don’t analyze how approximation errors accumulate layer-by-layer in deep transformer stacks.

3. The technical pipeline is quite dense, making it difficult for non-specialists to grasp. Doesn’t discuss latency or real-time trade-offs, which are critical for deployment.

**Questions:**

Could the spiking conversion harm adaptability or generalization on tasks beyond language modeling and reasoning?

---

> ### Author Response · Authors · 2025-11-25
> **Response to Reviewer 4kKu (1/3)**
>
> We greatly appreciate the reviewer’s feedback and suggestions. We respond to the questions asked by the reviewer individually. Key responses are `highlighted in the text`. We will incorporate the suggestions in the revised version.
>
> **Weakness 1: The window inhibition mechanism suppresses firing within a short temporal window to prevent over-firing, but this changes the temporal distribution of spikes. Since the quantizer and neuron are tightly coupled, any mismatch between inhibition strength and quantization scale could distort activation magnitude**.
>
> **[Answer for Weakness 1]**:
>
> **Although the introduction of window inhibition mechanism changes the temporal distribution of spikes, it will not affect the calculation results.** Due to that the window inhibition mechanism does not alter the charges accumulated upon neuron membrane, it does not distort activation strength. More specifically, we specify the detailed process of the window inhibition mechanism in Section A1. As illustrated in Algorithm 1, `for the equivalence between refined ST-BIF neuron with window inhibition mechanism and corresponding quantizer, we introduce a spike tracer` $S_{\rm com}$ `to trace the redundant spikes for compensation (e.g., the sum of spikes in the inhibiting window is greater than 1 or less than -1)`. After T time-step, $S_{\rm com}$ should `fire redundant spikes until there is no spike left to ensure the equivalence between ST-BIF neuron and corresponding quantizer`.
>
> We further randomly select 100 samples from WikiText2 benchmark to compare the mean square error (MSE) between activation (e.g., Query, Key, Value) output from quantizer and ST-BIF neuron under different inhibiting window lengths in R4-Table 1. As tabulated, although SpikingLLM without window inhibition mechanism (L=1) delivers the lowest mean square error and the best performance, the over-firing issue increases the energy consumption of converted SNNs compared to corresponding QLLMs. Instead, `the proposed window inhibition mechanism maintains consistent activation magnitude, effectively suppressing over-firing issues and significantly reducing the energy consumption with negligible performance degradation`. We will incorporate the experimental results and detailed analysis in the revised manuscript.
>
>      R4-Table 1: Comparison between activation (Query, Key, Value) output from quantizer and ST-BIF neuron under different inhibiting window lengths L.
>
> |    Method    | Category |    Model    |    Bits   | L |  Sparsity  | Energy(J) |    Query    |     Key     |    Value    | WikiText2 Perplexity | C4 Perplexity |
> |:------------:|:--------:|:-----------:|:---------:|:-:|:----------:|:---------:|:-----------:|:-----------:|:-----------:|:--------------------:|:-------------:|
> | PrefixQuant* |   QLLMs  |  Llama-2-7B | W4A4QKVS4 | - |      -     |    1.64   |      0.     |      0.     |      0.     |         11.56        |     14.10     |
> |  SpikingLLM  |   SNNs   |  Llama-2-7B | W4A4QKVS4 | 1 |   34.93%   |    1.70   | **2.48e-6** | **2.72e-6** | **3.06e-6** |       **10.91**      |   **13.68**   |
> |  SpikingLLM  |   SNNs   |  Llama-2-7B | W4A4QKVS4 | 2 |  _49.78%_  |   _1.19_  |  _2.78e-6_  |  _3.05e-6_  |  _3.42e-6_  |        _10.94_       |    _13.71_    |
> |  SpikingLLM  |   SNNs   |  Llama-2-7B | W4A4QKVS4 | 4 | **63.21%** |  **0.94** |   3.11e-6   |   3.32e-6   |   3.58e-6   |         10.99        |     13.78     |
> |------------|--------|-----------|---------|-|----------|---------|-----------|-----------|-----------|--------------------|-------------|
> | PrefixQuant* |   QLLMs  | Llama-2-13B | W4A4QKVS4 | - |      -     |    3.16   |      0.     |      0.     |      0.     |         7.99         |     10.68     |
> |  SpikingLLM  |   SNNs   | Llama-2-13B | W4A4QKVS4 | 1 |   32.82%   |    3.96   | **3.23e-6** | **3.18e-6** | **2.76e-6** |       **7.72**       |   **10.23**   |
> |  SpikingLLM  |   SNNs   | Llama-2-13B | W4A4QKVS4 | 2 |  _48.94%_  |   _2.66_  |  _3.45e-6_  |  _3.32e-6_  |  _2.91e-6_  |        _7.75_        |    _10.26_    |
> |  SpikingLLM  |   SNNs   | Llama-2-13B | W4A4QKVS4 | 4 | **62.51%** |  **2.08** |   3.79e-6   |   3.62e-6   |   3.12e-6   |         7.80         |     10.32     |

---

> ### Author Response · Authors · 2025-11-25
> **Response to Reviewer 4kKu (2/3)**
>
> **[Weakness 2] The authors use “differential strategies” from SpikeZIP-TF, but don’t analyze how approximation errors accumulate layer-by-layer in deep transformer stacks**.
>
> **[Answer for Weakness 2]**:
>
> Thanks for your insightful suggestions. As requested by reviewer, we randomly select 100 samples from WikiText2 benchmark to compare the mean square error (MSE) between QLLMs and SNNs with Llama-2-7B (8, 16, 24 and 32 layer) in R4-Table 2. Theoretically, differential strategies ensure the computation is event-driven and pipelined while does not introduce any approximation error, which is experimentally valiated by our experiment in R4-Table 2. Note that the minor computation difference is resulted from the numerical error of calculation, but will not affect the equivalence between SNNs and QLLMs. We will incorporate the experimental results and detailed analysis in the revised manuscript.
>
>       R4-Table 2: Comparison on layer-by-layer numerical error between QLLMs and SNNs with Llama-2-7B.
> |      Layer     |    8    |    16   |    24   |    32   |
> |:--------------:|:-------:|:-------:|:-------:|:-------:|
> | MSE (W4A4QKVS4) | 8.62e-6 | 9.04e-6 | 1.86e-5 | 2.38e-5 |
> | MSE (W4A5QKVS5) | 6.18e-6 | 7.32e-6 | 9.04e-6 | 1.28e-5 |
>
> **[Weakness 3] Lack of discussion on latency or real-time trade-offs, which are critical for deployment.**
>
> **[Answer for Weakness 3]**:
>
> As requested by the reviewer, we report the perplexity and energy over time-step with Llama-2-7B in R4-Table 3 and R4-Table 4. We then clarify that the inference latency is introduced by multi-timestep simulation in SpikingLLM, while the equilibrium time-step couples with the quantization level as tabulated in R4-Table 3 and R4-Table 4. Specifically, the equilibrium time-step of n-bit quantization is $2^{n}$(e.g., the time-step of Llama-2-7B W4A5QKVS5 is $2^{5}=32$ in R4-Table 4). `Note that as SNNs reach equilibrium, the energy also reaches a saturated state`. We will incorporate the experimental results and detailed analysis in the revised manuscript.
>
>      R4-Table 3: Perplexity and energy over time-step with Llama-2-7B W4A4QKVS4, the equilibrium perplexity is equal to Llama-2-7B W4A4QKVS4 (11.56).
> |   Time-step   |   8   |   9   |   10  |   11  |   12  |   13  |   14  |   15  |     16    |     17    |   18  |   19  |   20  |   30  |   40  |   50  |   60  |   70  |
> |:-------------:|:-----:|:-----:|:-----:|:-----:|:-----:|:-----:|:-----:|:-----:|:---------:|:---------:|:-----:|:-----:|:-----:|:-----:|:-----:|:-----:|:-----:|:-----:|
> | WikiText2 PPL | 21.28 | 17.05 | 14.49 | 12.94 | 11.99 | 11.44 | 11.16 | 11.04 | **10.99** | **10.99** | 11.00 | 11.01 | 11.03 | 11.24 | 11.38 | 11.48 | 11.56 | 11.56 |
> | Energy (J) | **0.47** | 0.53 | 0.59 | 0.64 | 0.71 | 0.78 | 0.83 | 0.88 | 0.94 | 0.97 | 0.99 | 1.01 | 1.03 | 1.07 | 1.11 | 1.13 | 1.15 | 1.15 |
>
>       R4-Table 4 Perplexity and energy over time-step with Llama-2-7B W4A5QKVS5, the equilibrium perplexity is equal to Llama-2-7B W4A5QKVS5 (7.78).
> |   Time-step   |   15  |   20  |  25  |  26  |  27  |  28  |  29  |  30  |  31  |    32    |    33    |    34    |    35    |  45  |  55  |  65  |  75  |  85  |
> |:-------------:|:-----:|:-----:|:----:|:----:|:----:|:----:|:----:|:----:|:----:|:--------:|:--------:|:--------:|:--------:|:----:|:----:|:----:|:----:|:----:|
> | WikiText2 PPL | 19.75 | 11.30 | 8.07 | 7.92 | 7.83 | 7.77 | 7.74 | 7.73 | 7.72 | **7.71** | **7.71** | **7.71** | **7.71** | 7.74 | 7.76 | 7.76 | 7.78 | 7.78 |
> | Energy (J) | **0.73** | 0.96 | 1.20 | 1.24 | 1.29 | 1.33 | 1.39 | 1.43 | 1.48 | 1.53 | 1.56 | 1.59 | 1.61 | 1.71 | 1.79 | 1.83 | 1.85 | 1.85 |

---

> ### Author Response · Authors · 2025-11-25
> **Response to Reviewer 4kKu (3/3)**
>
> **[Question 1] Could the spiking conversion harm adaptability or generalization on tasks beyond language modeling and reasoning?**
>
>
> **[Answer for Question 1]**:
>
>
>
> **The spiking conversion will not harm the adaptability or generalization on tasks beyond language modeling and reasoning**. The reasoning and language modeling tasks in our paper are oriented toward evaluating LLMs' ability to leverage contextual knowledge. Therefore, we conduct experiments on factual knowledge tasks(e.g., SciQ [1], BoolQ [2] and OpenBookQA [3]) that favor parameter-internal knowledge storage capability of LLMs to verify the adaptability of SpikingLLM. As tabulated in R4-Table 5, our SpikingLLM enables the equivalence between QLLMs and SNNs, achieving promising performance on factual knowledge tasks. `Notably, under the W4A5QKVS5 setting, SpikingLLM achieves comparable performance with FP16 Llama-2-13B, further demonstrating the adaptability of our method beyond language modeling and reasoning`. We will incorporate the experimental results and detailed analysis in the revised manuscript.
>
>             R4-Table 5: Comparison on Llama-2 models with factual knowledge tasks.
> |    Method    | Category | Fully-Spiking |  T |    Bits   |    SciQ   |   BoolQ   | OpenBookQA |    Avg.   |
> |:------------:|:--------:|:-------------:|:--:|:---------:|:---------:|:---------:|:----------:|:---------:|
> |  Llama-2-7B  |   ANNs   |       -       |  - |    FP16   |   91.00   |   77.71   |    44.20   |   70.97   |
> | PrefixQuant* |   QLLMs  |       -       |  - | W4A4QKVS4 |   86.70   |   68.23   |  **37.60** |   64.18   |
> |  SpikingLLM  |   SNNs   |       ✓       | 16 | W4A4QKVS4 | **86.80** | **68.31** |    37.50   | **64.20** |
> | PrefixQuant* |   QLLMs  |       -       |  - | W4A5QKVS5 |   89.10   |   72.45   |  **38.00** |   66.52   |
> |  SpikingLLM  |   SNNs   |       ✓       | 32 | W4A5QKVS5 | **89.20** | **72.51** |  **38.00** | **66.57** |
> |------------|--------|-------------|--|---------|---------|---------|----------|---------|
> |  Llama-2-13B |   ANNs   |       -       |  - |    FP16   |   93.50   |   80.55   |    45.20   |   73.08   |
> | PrefixQuant* |   QLLMs  |       -       |  - | W4A4QKVS4 |   91.10   |   75.44   |  **43.00** |   69.85   |
> |  SpikingLLM  |   SNNs   |       ✓       | 16 | W4A4QKVS4 | **91.30** | **75.62** |    42.90   | **69.94** |
> | PrefixQuant* |   QLLMs  |       -       |  - | W4A5QKVS5 |   92.50   |   79.54   |    44.20   |   72.08   |
> |  SpikingLLM  |   SNNs   |       ✓       | 32 | W4A5QKVS5 | **92.60** | **79.68** |  **44.30** | **72.19** |
>
> [**Reference**] \
> [1] J Welbl, et al. Crowdsourcing Multiple Choice Science Questions. \
> [2] C Clark, et al. BoolQ: Exploring the Surprising Difficulty of Natural Yes/No Questions. NAACL 2019 \
> [3] T Mihaylov, et al. Can a Suit of Armor Conduct Electricity? A New Dataset for Open Book Question Answering. EMNLP 2018

---

> ### Author Response · Authors · 2025-11-28
> **Sincerely looking forward to the further discussions**
>
> Dear reviewer,
>
> We are wondering if our response and revision have resolved your concerns. If our response has addressed your concerns, we would highly appreciate it if you could re-evaluate our work and consider raising the score.
>
> If you have any additional questions or suggestions, we would be happy to have further discussions.
>
> Best regards,
>
> The Authors

---

### Official Review · Reviewer_HRjH · 2025-11-01

**Soundness:** 2
**Presentation:** 3
**Contribution:** 2
**Rating:** 4
**Confidence:** 4

**Summary:**

This paper proposes a brain-inspired, energy-efficient framework for large language models that converts quantized LLMs into fully spiking neural networks. By introducing PrefixQuant* and mechanisms such as QK2Head-migration and window inhibition, the approach ensures all operators can be represented in spiking form while maintaining strong model performance. SpikingLLM effectively tackles SNN-unfriendly operations like KV Cache and Softmax, enabling complete equivalence between quantized and spiking models.

**Strengths:**

S1: This paper is in good written and easy to follow. The proposed method is simple, especially the Figure 2, which outlines the ANN-SNN conversion pipeline.
S2: The proposed method is mainly based on prefixQuant, which keeps the performance in low precision. This work may achieve somewhat energy efficiency according their metric.

**Weaknesses:**

W1: My main concern is the novelty of this work: both the ANN-SNN conversion and prefixQuant have already been studied. From my view, this work mainly replaces the quantization method in ANN-SNN conversion pipeline (like the mentioned SpikeZIP-TF). The discussions about the relationship between quantization and spiking neuron networks should go deeper.
W2: In the experiment, although the energy is very low compared with ANNs or the quantized LLM, it takes 16-32 times inference to simulate layer by layer. Therefore, from my view, this design is not suitable for LLMs.

**Questions:**

For the “inhibiting window”, how to ensure computational equivalence compared with quantization?

---

> ### Author Response · Authors · 2025-11-25
> **Response to Reviewer HRjH (1/5)**
>
> We greatly appreciate the reviewer’s feedback and suggestions. We respond to the questions asked by the reviewer individually. Key responses are `highlighted in the text`. We will incorporate the suggestions in the revised version.
>
> **Weakness 1: Both the ANN-SNN conversion and PrefixQuant have already been studied. From my view, this work mainly replaces the quantization method in ANN-SNN conversion pipeline (like the mentioned SpikeZIP-TF). The discussions about the relationship between quantization and spiking neuron networks should go deeper.**
>
> **[Answer for Weakness 1]**:
>
> We appreciate reviewer's understanding of ANN-to-SNN conversion flow, and that `the activation quantization plays an important role within ANN-to-SNN conversion pipeline`.
> Neverthless, we repsectively disagree with reviewer's comment about our innovation of this work.
>
> First of all, we want to highlight that this is the first work that successively `bridges the gap between SNN and pretrained LLM`, while simutaenously achieving high output quality, high energy efficiency and no-training overhead. Although we build our spikingLLM leveraging the neuron design and its conversion pipeline from SpikeZIP-TF [1], ahieving the aforementional objective is non-trivial that demands to address multiple critical challenges. **Our solutions summarized as follows are composed as methodological innovations in this work.**
>
>
> **Challenge-1：As tabulated in R3-Table 1, directly applying the Quantization-Aware Training method LSQ [2] in SpikeZIP-TF on LLMs fails to efficiently achieve effective performance. Meanwhile, simply applying the vanilla PrefixQuant [3] results in that the converted SNN contains non-spike communication and introduces inefficient computation.** More specifically, PrefixQuant neglects post-q and post-softmax quantization (i.e., cannot be further converted into the spike form) as shown in Figure 3, which fails to support the conversion of matrix product ($QK^{T}$, $\text{softmax}(\frac{QK^{T}}{\sqrt{d}})V$) into spike version, thus increasing the energy consumption of converted SNNs (as tabulated in R3-Table 2).
>
>
> **Our Solution: Rather than directly applying PrefixQuant, we implement the variant PrefixQuant\*,** we insert post-q and post-softmax quantization on the basis of PrefixQuant to introduce PrefixQuant* in section 4.1. `For 4-dimensional post-softmax output, we propose a novel strategy called QK2Head-migration quantization`, as tabulated in Figure 7, R3-Table 3 and R3-Table 4, directly applying original quantization in PrefixQuant to quantize softmax output (where km=1 and qm=1) apparently degrades the performance, `while our QK2Head-migration quantization (where km=16 and qm=1) significantly improves the post-softmax quantization performance`. Ultimately, the SNN converted from the QLLM using PrefixQuant* shows better energy efficiency compared to counterpart using prefixQuant.
>
>                         R3-Table 1: Comparison between LSQ and PrefixQuant* on Llama-2-7B.
> | Model         | Quantization Method | Quantization Type         |   GPU         |  Time         | Bits         | WikiText2 Perplexity |
> |:---------------:|:----------------------------------:|:--------------:|:--------------:|:--------------:|:--------------:|:----------------------:|
> | Llama-2-7B    | LSQ  | QAT                            |          4 NVIDIA-4090                  |            6 hours                | W4A4QKVS4    | 45.28                |
> | Llama-2-7B    | PrefixQuant* | PTQ        |         1 NVIDIA-4090               |           **269 seconds**             | W4A4QKVS4    | **11.56**               |
>
>         R3-Table 2: Comparison of energy consumption between PrefixQuant, PrefixQuant* and corresponding SNNs.
>  | Method         | Bits | post-q         |   post-softmax         |  2-7B energy(J)         | 2-7B SNNs energy(J)         | 2-13B energy(J)         | 2-13B  SNNs energy(J)  |
> |:---------------:|:----------------------------------:|:--------------:|:--------------:|:--------------:|:--------------:|:----------------------:|:----------------------:|
> | PrefixQuant    | W4A4KV4  | ✗   |          ✗                |            2.15               | 2.77(+0.62)    | 4.18                | 5.64(+1.46)              |
> | PrefixQuant*    | W4A4QKVS4  | ✓      |        ✓              |          1.64             | 0.94(-0.70)    | 3.16               |  2.08(-1.08)                |

---

> ### Author Response · Authors · 2025-11-25
> **Response to Reviewer HRjH (2/5)**
>
> **[Continue to the Answer for Weakness 1]**:
>
>      R3-Table 3: Wikitext2 perplexity with Llama-2-13B W4A4QKVS4 under various (km, qm) settings.
> | (km,qm)    | 1 | 4 | 16 | 64 |
> |:----:|:-------:|:----------:|:----------------:|:--------------:|
> | 1   | 42.31 | 31.06 | 14.77 | 13.10  |
> | 4    | 19.71 | 15.94 | 12.59 | 25.43  |
> | 16    | **7.99** | 10.02 | 22.74 | 138.62  |
> | 64   | 10.23 | 22.18 | 109.68 | 307.30 |
>
>       R3-Table 4: Wikitext2 perplexity with Llama-2-13B W4A5QKVS5 under various (km, qm) settings.
> | (km,qm)     | 1 | 4 | 16 | 64 |
> |:----:|:-------:|:----------:|:----------------:|:--------------:|
> | 1   | 8.96 | 7.45 | 6.46 | 8.88  |
> | 4    | 7.27 | 6.41 | 8.45 | 18.67  |
> | 16    | **6.28** | 8.19 | 17.08 | 105.49  |
> | 64   | 8.44 | 16.48 | 85.86 | 245.09 |
>
> **Challenge-2:  As tabulated in R3-Table 5, SpikeZIP-TF lacks strategies for converting LLMs-specific operators (e.g., KV Cache, RMSNorm) to their spike-equivalent version, which fails to enable the equivalence between SNNs and QLLMs.**
>
> **Our solution: We innovatively introduce Spike KV Cache(in Section 4.3), Spike SiLU, Spike RMSNorm and Spike Softmax (in Section A2).** `The introduction of spike operators above enables the equivalence between QLLMs and SNNs, which is experimentally verified in Table 4 and R3-Table 5.`
>
>                           R3-Table 5: Comparison between SpikingLLM and SpikeZIP-TF.
> |    Method   | Window Inhibition Mechanism | Spike Operators |    Bits   | Energy(J) | WikiText2 Perplexity | C4 Perplexity |
> |:-----------:|:---------------------------:|:---------------:|:---------:|:---------:|:--------------------:|:-------------:|
> | SpikeZIP-TF |           ✗                 |        ✗        | W4A4QKVS4 |    3.08   |        857.42        |     942.86    |
> |  SpikingLLM |           ✓                 |        ✓        | W4A4QKVS4 |  **0.94** |       **10.99**      |   **13.78**   |
>
> **Challenge-3: Simply applying the vanilla ST-BIF neuron (L=1 in R3-Table 6) introduces a great amount of redunant spike communication and computation, thus hampering the energy efficiency.** As ST-BIF neuron fires tenary spikes (+1,0,-1), it may existing repeatively generate +1 and -1 as neuron output (aka. spike oscillation), resulting in that the converted SNNs fail to reduce the energy consumption when converting the QLLMs to its SNN counterpart (3.16J->3.96J in R3-Table 6).
>
> **Our Solution: We propose a brain-inspired inihibition mechanism to mitigate such phenomenon.** As tabulated in Table 6 and R3-Table 6, with negligible performance degradation, our window inhibition mechanism effectively suppresses over-firing issues and significantly reduces the energy consumption of converted SNNs compared to corresponding QLLMs
>
>        R3-Table 6: Ablation of window inhibition mechanism on Llama-2-13B. L refers to inhibiting window length.
> |Method|Category| L         | Sparsity | Energy(J)         |   WikiText2 Perplexity        |  C4 Perplexity |
> |:---------------:|:---------------:|:---------------:|:----------------------------------:|:--------------:|:--------------:|:--------------:|
> | PrefixQuant* | QLLMs | -    | -  |         3.16                   |          7.99                  |            10.68                |
> | SpikingLLM | SNNs | 1    | 32.82%  |         3.96                   |          **7.72**                  |            **10.23**                |
> | SpikingLLM | SNNs | 2    | _48.94%_ |    _2.66_     |         _7.75_               |           _10.26_             |
> | SpikingLLM | SNNs | 4    | **62.51%** |   **2.08**      |         7.80              |           10.32             |
>
> **With the three solutions above, our SpikingLLM addresses the critical challenges on scaling ANN-to-SNN method up to LLMs, bridging the gap between SNNs and LLMs.**

---

> ### Author Response · Authors · 2025-11-25
> **Response to Reviewer HRjH (3/5)**
>
> **[Continue to the Answer for Weakness 1]**:
>
> Then we clarify the relationship between quantization and spiking neuron networks. We first highlight that the critical procedure of traditional ANN-to-SNN methods(e.g., SpikeZIP-TF [1], MST [4]) is to `replace the activation quantizer with equivalent spike neuron`, thus **converting the Multiply-ACcumulate operations (MACs) in to ACcumulate-Only operations (ACs).** Consequently, the activation quantization plays an important role within ANN-to-SNN (A2S) conversion pipeline. However, quantization methods that meet the following conditions are applicable to A2S.
>
> ①: **Since that the activation quantization plays an important role within ANN-to-SNN (A2S) conversion pipeline, weight-activation quantization methods are applicable to A2S**. Consequently, as tabulated in R3-Table 7, weight-only quantization method EfficientQAT [5] is not applicable to A2S.
>
> ②: **Instead of dynamic ones, static activation quantization methods are applicable to A2S**. In Section 3.2, we explain the reason for selecting static quantization method PrefixQuant rather than dynamic quantization method(e.g., OmniQuant [6]). Static quantization means that `the quantization scale of each quantizer is fixed at inference`, while `the quantization scale of each quantizer in dynamic quantization is dynamically determined by the input tensor at inference`. During SNNs inference, the threshold voltage $\vec{V}_{\rm thr}$ of neuron $\Theta$ is equal to the quantization scale $\vec{q}$ of corresponding quantizer. Consequently, `when tackling with the spiking version input(which means we can not acquire the total input before T time-step), static quantization with fixed quantization scale is more suitable to ANN-to-SNN method compared to dynamic quantization method where the quantization scale is dynamically determined by input`.
>
> ③: **Only static activation quantization methods encompassing all activations are applicable to A2S**. As illustrated in Challenge-1 of [**Answer for Weakness 1**], the vanilla PrefixQuant neglects post-q and post-softmax quantization, which fails to support the conversion of matrix product ($QK^{T}$, $\text{softmax}(\frac{QK^{T}}{\sqrt{d}})V$) into spike version, which is not applicable to A2S (as tabulated in R3-Table 7).
>
>                        R3-Table 7: Comparison of quantization methods on A2S.
> |    Method    | weight quantization | activation quantization | activation quantization type | quantization on all activations | applicable to A2S |
> |:------------:|:-------------------:|:-----------------------:|:----------------------------:|:-------------------------------:|:-----------------:|
> | EfficientQAT |          ✓          |            ✗            |               -              |                -                |         ✗         |
> |   OmniQuant  |          ✓          |            ✓            |            dynamic           |                ✗                |         ✗         |
> |  PrefixQuant |          ✓          |            ✓            |            static            |                ✗                |         ✗         |
> | PrefixQuant* |          ✓          |            ✓            |            static            |                ✓                |         ✓         |
>
> As illustrated in solution for challenge-1 of [**Answer for Weakness 1**], we `introduce a novel QK2Head-migration post-softmax quantization alongwith post-q quantization on the basis of PrefixQuant to introduce PrefixQuant* in section 4.1`. Our PrefixQuant* meets the three conditions above, which is applicable to ANN-to-SNN conversion pipeline.

---

> ### Author Response · Authors · 2025-11-25
> **Response to Reviewer HRjH (4/5)**
>
> **[Continue to the Answer for Weakness 1]**:
>
> To verify the generalization ability of SpikingLLM, we further compare SpikingLLM with different static QLLMs on Llama-2-7B in R3-Table 8. For OmniQuant [6], we replace the dynamic activation quantizers with static activation quantizers to establish OmniQuant\*. For weight-only quantization method EfficientQAT [5], we incorporate static activation quantizers similar to PrefixQuant\* in Figure 3 to establish EfficientQAT\*. `As tabulated, our SpikingLLM enables the equivalence between the static QLLMs and SNNs, verifying the effectiveness of SpikingLLM on arbitrary static QLLMs`. We will incorporate the experimental results and detailed analysis in the revised manuscript.
>
>                 R3-Table 8: Comparison between SpikingLLM and different static QLLMs on Llama-2-7B.
> | Method            | Category        | bits    | Time-step    |  WikiText2 Perplexity    | PIQA | ARC-e | ARC-c | HellaSwag | Winogrand | Avg.  |
> |:-----------------:|:---------------------:|:----------:|:----------:|:-----------:|:-----------:|:----:|:-----:|:-----:|:---------:|:---------:|
> | OmniQuant\*  | static QLLMs  | W4A4QKVS4       |      -     |    12.49     | 70.9 | 61.2  | **35.8**  | 66.9      | **60.8**      | 59.12 |
> | SpikingLLM | SNNs  | W4A4QKVS4       |       16     |   **12.32**      | **71.1** | **61.4**  | 35.6  | **67.0**      | 60.6    | **59.14** |
> |-----------------|---------------------|----------|----------|-----------|-----------|----|-----|-----|---------|---------|-----|
> | EfficientQAT\* | static QLLMs  | W4A4QKVS4  | -    |    11.32      |     72.9      | 63.1 | **36.8**  | 69.2 | **61.8**      | 60.76 |
> | SpikingLLM | SNNs  | W4A4QKVS4       |     16     |     **10.78**      | **73.0** | **63.2**  | 36.7  | **69.3**      | **61.8**      | **60.80** |
>
> **Weakness 2: In the experiment, although the energy is very low compared with ANNs or the quantized LLM, it takes 16-32 times inference to simulate layer by layer. Therefore, from my view, this design is not suitable for LLMs.**
>
> **[Answer for Weakness 2]**:
>
> We appreciate the reviewer’s attention to the inference latency introduced by multi-timestep simulation in SpikingLLM. However, we first highlight that our SpikingLLM is **simulated step-by-step, rather than layer-by-layer**. SpikingJelly [7] `specifies that the step-by-step pattern is closer to the behavior of most event-driven neuromorphic chips(e.g., Intel Loihi [8], SpiNNaker [9]) compared to layer-by-layer, verifying the accessibility of step-by-step inference on event-driven neuromorphic chips`. We then claim that **SpikingLLM is advantageous for deploying LLMs on edge devices**, for the following reasons:
>
> **① Energy Efficiency is the Primary Bottleneck for Edge LLMs**: As highlighted in [10], the deployment of LLMs on resource-constrained devices is severely limited by computational and memory constraints, with energy consumption being a dominant factor. Instead, our SpikingLLM achieves a `drastic reduction in energy consumption` (e.g., 0.94J for SpikingLLM vs. 1.64J for QLLMs in R3-Table 2) by replacing Multiply-Accumulate (MAC) operations with Accumulate-Only (AC) operations, which demonstrates outstanding energy-efficiency.
>
> **② Multi-timestep Overhead is Mitigated by Hardware and Sparsity**: Neuromorphic and spiking-friendly hardware (e.g., Intel Loihi [8], SpiNNaker [9]) can execute multi-timestep in parallel or in a pipelined manner, effectively reducing latency. While the neuromorphic hardwares above support the step-by-step inference [7]. Moreover, as tabulated in Table 6 and R3-Table 3, the introduction of window inhibition mechanism increases the sparsity of SpikingLLM by up to 60%, reducing the number of active operations per timestep and further lowering energy. Consequently, the multi-timestep overhead can be mitigated by hardware support and high sparsity of SpikingLLM.
>
> Although the multi-timestep inference of SpikingLLM introduces latency, it enables unprecedented energy savings, which is critical for on-device LLMs deployment. We will incorporate the detailed analysis in the revised manuscript.

---

> ### Author Response · Authors · 2025-11-25
> **Response to Reviewer HRjH (5/5)**
>
> **Question 1: For the “inhibiting window”, how to ensure computational equivalence compared with quantization?**
>
> **[Answer for Question 1]**:
>
> Thanks for your insightful question. In Section A1, we specify the detailed process of the window inhibition mechanism. As illustrated in Algorithm 1, we introduce an inhibiting window with length L to combine the original L time-step spikes into 1 time-step spike. `For the equivalence between refined ST-BIF neuron with window inhibition mechanism and corresponding quantizer, we introduce a spike tracer` $S_{\rm com}$ `to trace the redundant spikes for compensation (e.g., the sum of spikes in the inhibiting window is greater than 1 or less than -1)`. After T time-step, $S_{\rm com}$ should `fire redundant spikes until there is no spike left to ensure the equivalence between ST-BIF neuron and corresponding quantizer`.
>
> We further randomly select 100 samples from WikiText2 benchmark to compare the mean square error between activation (e.g., Query, Key, Value) output from quantizer and ST-BIF neuron under different inhibiting window lengths in R3-Table 9. As tabulated, although SpikingLLM without window inhibition mechanism (L=1) delivers the lowest mean square error and the best performance, the over-firing issue increases the energy consumption of converted SNNs compared to corresponding QLLMs. Instead, `the proposed window inhibition mechanism maintains consistent activation magnitude, effectively suppressing over-firing issues and significantly reducing the energy consumption with negligible performance degradation`. We will incorporate the experimental results and detailed analysis in the revised manuscript.
>
>      R3-Table 9: Comparison between activation (Query, Key, Value) output from quantizer and ST-BIF neuron under different inhibiting window lengths L.
> |    Method    | Category |    Model    |    Bits   | L |  Sparsity  | Energy(J) |    Query    |     Key     |    Value    | WikiText2 Perplexity | C4 Perplexity |
> |:------------:|:--------:|:-----------:|:---------:|:-:|:----------:|:---------:|:-----------:|:-----------:|:-----------:|:--------------------:|:-------------:|
> | PrefixQuant* |   QLLMs  |  Llama-2-7B | W4A4QKVS4 | - |      -     |    1.64   |      0.     |      0.     |      0.     |         11.56        |     14.10     |
> |  SpikingLLM  |   SNNs   |  Llama-2-7B | W4A4QKVS4 | 1 |   34.93%   |    1.70   | **2.48e-6** | **2.72e-6** | **3.06e-6** |       **10.91**      |   **13.68**   |
> |  SpikingLLM  |   SNNs   |  Llama-2-7B | W4A4QKVS4 | 2 |  _49.78%_  |   _1.19_  |  _2.78e-6_  |  _3.05e-6_  |  _3.42e-6_  |        _10.94_       |    _13.71_    |
> |  SpikingLLM  |   SNNs   |  Llama-2-7B | W4A4QKVS4 | 4 | **63.21%** |  **0.94** |   3.11e-6   |   3.32e-6   |   3.58e-6   |         10.99        |     13.78     |
> |------------|--------|-----------|---------|-|----------|---------|-----------|-----------|-----------|--------------------|-------------|
> | PrefixQuant* |   QLLMs  | Llama-2-13B | W4A4QKVS4 | - |      -     |    3.16   |      0.     |      0.     |      0.     |         7.99         |     10.68     |
> |  SpikingLLM  |   SNNs   | Llama-2-13B | W4A4QKVS4 | 1 |   32.82%   |    3.96   | **3.23e-6** | **3.18e-6** | **2.76e-6** |       **7.72**       |   **10.23**   |
> |  SpikingLLM  |   SNNs   | Llama-2-13B | W4A4QKVS4 | 2 |  _48.94%_  |   _2.66_  |  _3.45e-6_  |  _3.32e-6_  |  _2.91e-6_  |        _7.75_        |    _10.26_    |
> |  SpikingLLM  |   SNNs   | Llama-2-13B | W4A4QKVS4 | 4 | **62.51%** |  **2.08** |   3.79e-6   |   3.62e-6   |   3.12e-6   |         7.80         |     10.32     |
>
> [**Reference**] \
> [1] K You, et al. SpikeZIP-TF: Conversion is All You Need for Transformer-based SNN. ICML 2024 \
> [2] Steven K. Esser, et al. Learned Step Size Quantization. ICLR 2020 \
> [3] M Zhao, et al. PrefixQuant: Eliminating Outliers by Prefixed Tokens for Large Language Models Quantization. \
> [4] Z Wang, et al. Masked Spiking Transformer. ICCV 2023 \
> [5] M Zhao, et al. Efficient quantization-aware training for large language models. ACL main 2025 \
> [6] W Shao, et al. Omniquant: Omnidirectionally calibrated quantization for large language models. ICLR 2024 \
> [7] W Fang, et al. SpikingJelly: An open-source machine learning infrastructure platform for spike-based intelligence. Science Advances 2024 \
> [8] M Davies, et al. Loihi: A Neuromorphic Manycore Processor with On-Chip Learning. \
> [9] E Painkras, et al. SpiNNaker: A 1-W 18-Core System-on-Chip for Massively-Parallel Neural Network Simulation. \
> [10] Y Zheng, et al. A Review on Edge Large Language Models: Design, Execution, and Applications. ACM Computing Surveys 2024

---

> ### Author Response · Authors · 2025-11-28
> **Sincerely looking forward to the further discussions**
>
> Dear reviewer,
>
> We are wondering if our response and revision have resolved your concerns. If our response has addressed your concerns, we would highly appreciate it if you could re-evaluate our work and consider raising the score.
>
> If you have any additional questions or suggestions, we would be happy to have further discussions.
>
> Best regards,
>
> The Authors

---

### Official Review · Reviewer_mvGP · 2025-11-01

**Soundness:** 2
**Presentation:** 3
**Contribution:** 2
**Rating:** 4
**Confidence:** 4

**Summary:**

This study builds upon the PrefixQuant framework and introduces a novel unstructured, spike-driven inference method. In contrast to conventional quantization techniques, it employs an ANN-to-SNN conversion approach to enable spiking-based inference during the model’s inference phase. Moreover, it utilizes three quantization levels (-1, 0, 1) for spike encoding, aligning with methods like SpikeZIP-TF and SpikeLM. Experimental results demonstrate that the proposed approach surpasses both PrefixQuant and SpikeLLM in terms of efficiency.

**Strengths:**

1) The framework primarily relies on an ANN-SNN conversion pipeline, which is applied after training. As a result, the training cost is relatively lower compared to training models from scratch.

2) The energy efficiency and accuracy achieved are comparable to, or even surpass, those of quantization-based methods that incorporate spiking neuronal dynamics.

3) This research addresses a crucial area within the SNN domain. Advancing more efficient spiking large language models or improving ANN-SNN conversion techniques is essential for progress in this field.

**Weaknesses:**

1) From my view, what this work advances is not addressed, because there are may other better quantization methods, why to select PrefixQuant is not addressed.

2) If this work focuses on energy efficient LLMs, why not directly apply additive neural networks, such as BitNet and BitNet-1.58bit? when using 4-bit activations, the performance could be further compared.

**Questions:**

The comparison with other ANN-SNN conversion method could be addressed, even if comparing in small scale.

---

> ### Author Response · Authors · 2025-11-25
> **Response to Reviewer mvGP (1/4)**
>
> We greatly appreciate the reviewer’s feedback and suggestions. We respond to the questions asked by the reviewer individually. Key responses are `highlighted in the text`. We will incorporate the suggestions in the revised version.
>
> **Weakness 1: From my view, what this work advances is not addressed, because there are may other better quantization methods, why to select PrefixQuant is not addressed.**
>
> **[Answer for Weakness 1]**:
>
> Thanks for your insightful suggestion. We first highlight that the critical procedure of traditional ANN-to-SNN methods(e.g., SpikeZIP-TF [1], MST [2]) is to `replace the activation quantizer with equivalent spike neuron`, thus **converting the Multiply-ACcumulate operations (MACs) in to ACcumulate-Only operations (ACs).** Consequently, the activation quantization plays an important role within ANN-to-SNN (A2S) conversion pipeline. However, quantization methods that meet the following conditions are applicable to A2S.
>
> ①: **Since that the activation quantization plays an important role within ANN-to-SNN (A2S) conversion pipeline, weight-activation quantization methods are applicable to A2S**. Consequently, as tabulated in R2-Table 1, weight-only quantization method EfficientQAT [3] is not applicable to A2S.
>
> ②: **Instead of dynamic ones, static activation quantization methods are applicable to A2S**. In Section 3.2, we explain the reason for selecting static quantization method PrefixQuant [4] rather than dynamic quantization method(e.g., OmniQuant [5]). Static quantization means that `the quantization scale of each quantizer is fixed at inference`, while `the quantization scale of each quantizer in dynamic quantization is dynamically determined by the input tensor at inference`. During SNNs inference, the threshold voltage $\vec{V}_{\rm thr}$ of neuron $\Theta$ is equal to the quantization scale $\vec{q}$ of corresponding quantizer. Consequently, `when tackling with the spiking version input(which means we can not acquire the total input before T time-step), static quantization with fixed quantization scale is more suitable to ANN-to-SNN method compared to dynamic quantization method where the quantization scale is dynamically determined by input`.
>
> ③: **Only static activation quantization methods encompassing all activations are applicable to A2S**. As illustrated in R2-Table 2, the vanilla PrefixQuant neglects post-q and post-softmax quantization, which fails to support the conversion of matrix product ($QK^{T}$, $\text{softmax}(\frac{QK^{T}}{\sqrt{d}})V$) into spike version, which increases the energy consumption of converted SNNs and is not applicable to A2S.
>
>                       R2-Table 1: Comparison of quantization methods on A2S.
> |    Method    | weight quantization | activation quantization | activation quantization type | quantization on all activations | applicable to A2S |
> |:------------:|:-------------------:|:-----------------------:|:----------------------------:|:-------------------------------:|:-----------------:|
> | EfficientQAT |          ✓          |            ✗            |               -              |                -                |         ✗         |
> |   OmniQuant  |          ✓          |            ✓            |            dynamic           |                ✗                |         ✗         |
> |  PrefixQuant |          ✓          |            ✓            |            static            |                ✗                |         ✗         |
> | PrefixQuant* |          ✓          |            ✓            |            static            |                ✓                |         ✓         |
>
>         R2-Table 2: Comparison of energy consumption between PrefixQuant, PrefixQuant* and corresponding SNNs.
>  | Method         | Bits | post-q         |   post-softmax         |  2-7B energy(J)         | 2-7B SNNs energy(J)         | 2-13B energy(J)         | 2-13B  SNNs energy(J)  |
> |:---------------:|:----------------------------------:|:--------------:|:--------------:|:--------------:|:--------------:|:----------------------:|:----------------------:|
> | PrefixQuant    | W4A4KV4  | ✗   |          ✗                |            2.15               | 2.77 (+0.62)    | 4.18                | 5.64 (+1.46)              |
> | PrefixQuant*    | W4A4QKVS4  | ✓      |        ✓              |          1.64             | 0.94 (-0.70)    | 3.16               |  2.08 (-1.08)                |
>
> In this paper we `introduce a novel QK2Head-migration post-softmax quantization alongwith post-q quantization on the basis of PrefixQuant to introduce PrefixQuant* in section 4.1`. Our PrefixQuant* meets the three conditions above, which is applicable to ANN-to-SNN conversion pipeline.

---

> ### Author Response · Authors · 2025-11-25
> **Response to Reviewer mvGP (2/4)**
>
> **[Continue to the Answer for Weakness 1]**:
>
> To verify the generalization ability of SpikingLLM, we further compare SpikingLLM with different static QLLMs on Llama-2-7B in R2-Table 3. For OmniQuant [5], we replace the dynamic activation quantizers with static activation quantizers to establish OmniQuant\*. For weight-only quantization method EfficientQAT [3], we incorporate static activation quantizers similar to PrefixQuant\* in Figure 3 to establish EfficientQAT\*. `As tabulated, our SpikingLLM enables the equivalence between the static QLLMs and SNNs, verifying the effectiveness of SpikingLLM on arbitrary static QLLMs`. We will incorporate the the experimental results and detailed analysis in the revised manuscript.
>
>                    R2-Table 3: Comparison between SpikingLLM and different static QLLMs on Llama-2-7B.
> | Method            | Category        | bits    | Time-step    |  WikiText2 Perplexity    | PIQA | ARC-e | ARC-c | HellaSwag | Winogrand | Avg.  |
> |:-----------------:|:---------------------:|:----------:|:----------:|:-----------:|:-----------:|:----:|:-----:|:-----:|:---------:|:---------:|
> | OmniQuant\*  | static QLLMs  | W4A4QKVS4       | -  |    12.49     | 70.9 | 61.2  | **35.8**  | 66.9      | **60.8**      | 59.12 |
> | SpikingLLM | SNNs  | W4A4QKVS4       |       16     |   **12.32**      | **71.1** | **61.4**  | 35.6  | **67.0**      | 60.6    | **59.14** |
> |-----------------|---------------------|----------|----------|-----------|-----------|----|-----|-----|---------|---------|-----|
> | EfficientQAT\* | static QLLMs  | W4A4QKVS4  | -    |    11.32      |     72.9      | 63.1 | **36.8**  | 69.2 | **61.8**      | 60.76 |
> | SpikingLLM | SNNs  | W4A4QKVS4       | 16 | **10.78** | **73.0** | **63.2**  | 36.7  | **69.3**      | **61.8**      | **60.80** |
>
> **Weakness 2: If this work focuses on energy efficient LLMs, why not directly apply additive neural networks, such as BitNet and BitNet-1.58bit? The performance could be further compared.**
>
> **[Answer for Weakness 2]**:
>
> Thanks for your insightful suggestions. We respond to the question with the following points:
>
> ①: We first emphasize the difference between SpikingLLM and BitNet-1.58bit [6]. BitNet-1.58bit mainly focuses on weight quantization, leveraging ternary values({-1, 0, 1}) to quantize LLM parameter and improving energy efficiency. While as illustrated in [**Answer for Weakness 1**], our SpikingLLM `mainly focus on effective activation quantization LLMs applicable to ANN-to-SNN method`. Low bit-width quantization method can be applied on our SNN, as it is orthogonal to our current ANN-to-SNN conversion pipeline. However, compressing weights in lower bit-width is not our focus in this work.
>
> ②: Reducing the energy cost for activation communication plays an more important role in low energy AI inference engine [7,8]. For neurmorphic computing hardware [9,10], with dedicated in-memory computing and dataflow computing architecture, weight access can be minimized due to in-site access for compute but data movement of activation cannot be avoided. The communication cost significantly grows when model size further grows. Neverthless, BitNet-1.58bit is trained from scratch with 1.58-bit weights and 8-bit activations, **the $\textit{Activation-Activation}$ ($\textit{aka}$. AA) matrix products in the attention layer remain 8-bit calculations**, which is energy-inefficient.
>
> ③: We conduct experiments comparing SpikingLLM with other efficient LLMs similar to BitNet-1.58bit, such as MatMul-free LLM [11] and ShiftAddLLM [12], in Section A6. Experimental results further verify the superiority of SpikingLLM. We also conduct a comprehensive comparison between SpikingLLM and other efficient LLMs from various aspects in R2-Table 4, **with comparable model parameters, our SpikingLLM efficiently achieves better performance against Matmul-free LLM and BitNet-1.58bit**. We will incorporate the the experimental results and detailed analysis in the revised manuscript.
>
>       R2-Table 4: Comparison between SpikingLLM, Matmul-free LLM and BitNet-1.58bit on zero-shot common scene reasoning tasks.
> | Method            | Model                 | Parameters | MatMul-free | bits        | PIQA | ARC-e | ARC-c | HellaSwag | Winogrand | Avg.  | GPU            | Time          |
> |:-----------------:|:---------------------:|:----------:|:-----------:|:-----------:|:----:|:-----:|:-----:|:---------:|:---------:|:-----:|:---------:|:-----:|
> | BitNet b1.58  | Llama-1.3B  | 1.3B       | ✗  | W1.58A8QKV8          | **70.0** | _54.9_  | 24.2  | 37.7      | **55.8**      | 48.52 | 64 NVIDIA-H100 | 36 hours |
> | MatMul-free LLM   | MatMul-free-LLM-1.3B  | 1.3B       | ✓| ——          | 68.4 | 54.0  | _25.9_  | _44.9_      | 52.4 | _49.12_ | 8 NVIDIA-H100  | 84 hours      |
> | SpikingLLM(ours)        | Llama-3.2-1B  | 1.0B         | ✓ | W4A4QKVS4   | _68.6_ | **58.2**  | **29.9**  | **55.6**      | _53.8_  | **53.22** | 1 NVIDIA-4090  | **72 seconds**   |

---

> ### Author Response · Authors · 2025-11-25
> **Response to Reviewer mvGP (3/4)**
>
> **Question 1: The comparison with other ANN-SNN conversion method could be addressed, even if comparing in small scale.**
>
> **[Answer for Question 1]**:
>
> Thanks for your insightful suggestions. So far, SpikeLLM [13] delivers the state-of-the-art performance on scaling SNNs up to LLMs, even though the converted SNNs are not fully spiking. In Table 2 and Table 3, we select SpikeLLM as the baseline of our SpikingLLM.
> `Although ANN-SNN conversion methods, such as SpikeZIP-TF [1] and MST [2], deliver promising performance on Transformer-based models, they face several critical difficulties on scaling up to LLMs`:
>
>
> **Challenge-1：As tabulated in R2-Table 5, directly applying the Quantization-Aware Training method LSQ [14] in SpikeZIP-TF on LLMs fails to efficiently achieve effective performance.**
>
> **Our Solution: We insert post-q and post-softmax quantization on the basis of PrefixQuant to introduce PrefixQuant\* in section 4.1.** As tabulated in R2-Table 5, compared to LSQ in SpikeZIP-TF, our PrefixQuant* `efficiently achieves competitive performance on QLLMs`.
>
>                R2-Table 5: Comparison between LSQ and PrefixQuant* on Llama-2-7B.
> | Model         | Quantization Method | Quantization Type         |   GPU         |  Time         | Bits         | WikiText2 Perplexity | C4 Perplexity |
> |:---------------:|:----------------------------------:|:--------------:|:--------------:|:--------------:|:--------------:|:----------------------:|:----------------------:|
> | Llama-2-7B    | LSQ  | QAT                            |          4 NVIDIA-4090                  |            6 hours                | W4A4QKVS4    | 45.28                | 60.78                |
> | Llama-2-7B    | PrefixQuant* | PTQ        |         1 NVIDIA-4090               |           **269 seconds**             | W4A4QKVS4    | **11.56**               |      **14.10**               |
>
> **Challenge-2:  As tabulated in R2-Table 6, SpikeZIP-TF lacks strategies for converting LLMs-specific operators(e.g., KV Cache, RMSNorm) to their spike-equivalent version, which fails to enable the equivalence between SNNs and QLLMs.**
>
> **Our solution: We innovatively introduce Spike KV Cache(in Section 4.3), Spike SiLU, Spike RMSNorm and Spike Softmax (in Section A2).** `The introduction of spike operators above enables the equivalence between QLLMs and SNNs, which is experimentally verified in Table 4 and R2-Table 6.`
>
>
>                   R2-Table 6: Comparison between SpikingLLM and SpikeZIP-TF.
> |    Method   | Window Inhibition Mechanism | Spike Operators |    Bits   | Energy(J) | WikiText2 Perplexity | C4 Perplexity |
> |:-----------:|:---------------------------:|:---------------:|:---------:|:---------:|:--------------------:|:-------------:|
> | SpikeZIP-TF |           ✗                 |        ✗        | W4A4QKVS4 |    3.08   |        857.42        |     942.86    |
> |  SpikingLLM |           ✓                 |        ✓        | W4A4QKVS4 |  **0.94** |       **10.99**      |   **13.78**   |
>
>
> **Challenge-3: Simply applying the vanilla ST-BIF neuron (L=1 in R2-Table 7) introduces a great amount of redunant spike communication and computation, thus hampering the energy efficiency.** As ST-BIF neuron fires tenary spikes (+1,0,-1), it may existing repeatively generate +1 and -1 as neuron output (aka. spike oscillation), resulting in that the converted SNNs fail to reduce the energy consumption when converting the QLLMs to its SNN counterpart (3.16J->3.96J in R1-Table 6).
>
> **Our Solution: We propose a brain-inspired inihibition mechanism to mitigate such phenomenon.** As tabulated in Table 6 and R2-Table 7, with negligible performance degradation, our window inhibition mechanism effectively suppresses over-firing issues and significantly reduces the energy consumption of converted SNNs compared to corresponding QLLMs
>
>        R2-Table 7: Ablation of window inhibition mechanism on Llama-2-13B. L refers to inhibiting window length.
> |Method|Category| L         | Sparsity | Energy(J)         |   WikiText2 Perplexity        |  C4 Perplexity |
> |:---------------:|:---------------:|:---------------:|:----------------------------------:|:--------------:|:--------------:|:--------------:|
> | PrefixQuant* | QLLMs | -    | -  |         3.16                   |          7.99                  |            10.68                |
> | SpikingLLM | SNNs | 1    | 32.82%  |         3.96                   |          **7.72**                  |            **10.23**                |
> | SpikingLLM | SNNs | 2    | _48.94%_ |    _2.66_     |         _7.75_               |           _10.26_             |
> | SpikingLLM | SNNs | 4    | **62.51%** |   **2.08**      |         7.80              |           10.32             |
>
> **With the three solutions above, our SpikingLLM addresses the critical challenges on scaling ANN-to-SNN method up to LLMs, bridging the gap between SNNs and LLMs.**

---

> ### Author Response · Authors · 2025-11-25
> **Response to Reviewer mvGP (4/4)**
>
> **[Continue to the Answer for Question 1]**:
>
> We further conduct comparison between SpikeZIP-TF, SpikeLLM and SpikingLLM on Llama-2-7B in R2-Table 8. With the experimental results in R2-Table 8 and three solutions concluded above, **our SpikingLLM addresses the critical challenges on scaling ANN-to-SNN method up to LLMs, bridging the gap between SNNs and LLMs while significantly reducing the energy consumption**. We will incorporate the the experimental results and detailed analysis in the revised manuscript.
>
>      R2-Table 8: Comparison between SpikeZIP-TF, SpikeLLM and SpikingLLM on Llama-2-7B.
> |    Method   | Fully-Spiking |    Bits   | Energy(J) | WikiText2 Perplexity | C4 Perplexity |
> |:-----------:|:-------------:|:---------:|:---------:|:--------------------:|:-------------:|
> | SpikeZIP-TF |       ✓       | W4A4QKVS4 |    3.08   |        857.42        |     942.86    |
> |   SpikeLLM  |       ✗       |  W4A4KV4  |    5.18   |         11.46        |     14.45     |
> |  SpikingLLM(ours) |       ✓       | W4A4QKVS4 |  **0.94** |       **10.99**      |   **13.78**   |
>
> [**Reference**] \
> [1] K You, et al. SpikeZIP-TF: Conversion is All You Need for Transformer-based SNN. ICML 2024 \
> [2] Z Wang, et al. Masked Spiking Transformer. ICCV 2023 \
> [3] M Zhao, et al. Efficient quantization-aware training for large language models. ACL main 2025 \
> [4] M Zhao, et al. PrefixQuant: Eliminating Outliers by Prefixed Tokens for Large Language Models Quantization. \
> [5] W Shao, et al. Omniquant: Omnidirectionally calibrated quantization for large language models. ICLR 2024 \
> [6] S Ma, et al. The Era of 1-bit LLMs: All Large Language Models are in 1.58 Bits. \
> [7] YH Chen, et al. Eyeriss: An Energy-Efficient Reconfigurable Accelerator for Deep Convolutional Neural Networks. IEEE Journal of Solid-State Circuits 2017 \
> [8] W Wan, et al. A compute-in-memory chip based on resistive random-access memory. Nature 2022 \
> [9] M Davies, et al. Loihi: A Neuromorphic Manycore Processor with On-Chip Learning. \
> [10] E Painkras, et al. SpiNNaker: A 1-W 18-Core System-on-Chip for Massively-Parallel Neural Network Simulation. \
> [11] R Zhu, et al. Scalable MatMul-free Language Modeling. \
> [12] H You, et al. Shiftaddllm: Accelerating pretrained llms via post-training multiplication-less reparameterization. \
> [13] X Xing, et al. SpikeLLM: Scaling up Spiking Neural Network to Large Language Models via Saliency-based Spiking. ICLR 2025\
> [14] Steven K. Esser, et al. Learned Step Size Quantization. ICLR 2020

---

> ### Author Response · Authors · 2025-11-28
> **Sincerely looking forward to the further discussions**
>
> Dear reviewer,
>
> We are wondering if our response and revision have resolved your concerns. If our response has addressed your concerns, we would highly appreciate it if you could re-evaluate our work and consider raising the score.
>
> If you have any additional questions or suggestions, we would be happy to have further discussions.
>
> Best regards,
>
> The Authors

---

### Official Review · Reviewer_DZKi · 2025-11-01

**Soundness:** 2
**Presentation:** 3
**Contribution:** 2
**Rating:** 4
**Confidence:** 3

**Summary:**

This paper introduces SpikingLLM, a fully spiking version of quantized large language models (QLLMs) designed for energy-efficient inference. The method achieves equivalence between QLLMs and spiking neural networks (SNNs) through a QK2Head-migration post-softmax quantization and differential-based handling of SNN-unfriendly operators like KV Cache. It further refines the ST-BIF+ neuron with a window inhibition mechanism, effectively mitigating overfiring and enhancing sparsity. Experimental results show that SpikingLLM significantly reduces energy consumption while achieving state-of-the-art perplexity and reasoning performance, outperforming prior A2S (ANN-to-SNN) and DT (direct training) approaches.

**Strengths:**

1. This paper combines spiking neural networks (SNNs) and large language models (LLMs) to propose SpikingLLM, which exhibits certain biological neuron interpretability and low-power characteristics. This work focuses on the interesting topic of spiking large language models.
2. The paper obtains SNNs through a conversion-based quantization method, enabling the training of spiking-driven models with only a short amount of computation time.
3. The paper employs ternary spike representations, extending the precision-fitting capability of spike-based computation and enhancing the model’s expressiveness. However, it lacks rigorous comparative experiments, and the authors are encouraged to discuss this aspect in more depth.

**Weaknesses:**

1. The main weakness of this paper is that it does not address a clearly defined problem in either the SNN or quantization domains. The authors use PrefixQuant to improve SNN conversion performance through its strong quantization effects. However, the contribution to the field is limited: first, the ANN-to-SNN approach is a conventional method for obtaining SNNs; second, using ternary neuron operators to fit quantized or full-precision LLMs has already been explored in SpikeZIP-TF; and finally, if the performance gain mainly comes from PrefixQuant rather than methodological innovation, the novelty is insufficient.
2. Building on the first point, the paper lacks a rigorous validation of the proposed method’s effectiveness and generalization. For example, in Tables 2 and 3, the authors mainly compare with PrefixQuant, without showing whether the method generalizes to other quantization approaches. Therefore, it is suggested that the authors strengthen the discussion and contribution of the ANN-to-SNN conversion method itself, rather than relying heavily on a specific quantization technique.

**Questions:**

Please see my comments provided above.

---

> ### Author Response · Authors · 2025-11-25
> **Response to Reviewer DZKi (1/4)**
>
> We greatly appreciate the reviewer’s feedback and suggestions. We respond to the questions asked by the reviewer individually. Key responses are `highlighted in the text`. We will incorporate the suggestions in the revised version.
>
> **Weakness1: The main weakness of this paper is that it does not address a clearly defined problem in either the SNN or quantization domains. However, the contribution to the field is limited: first, the ANN-to-SNN approach is a conventional method for obtaining SNNs; second, using ternary neuron operators to fit quantized or full-precision LLMs has already been explored in SpikeZIP-TF; and finally, if the performance gain mainly comes from PrefixQuant rather than methodological innovation, the novelty is insufficient.**
>
> **[Answer for Weakness 1]**:
>
> We appreciate reviewer's understanding of the ANN-to-SNN conversion flow, and that the activation quantization plays an important role within ANN-to-SNN conversion pipeline.
> Neverthless, we repsectively disagree with reviewer's comment about our innovation of this work.
>
> First of all, we want to highlight that this is the first work that successively `bridges the gap between SNN and pretrained LLM`, while simutaenously achieving high output quality, high energy efficiency and no-training overhead. Although we build our spikingLLM leveraging the neuron design and its conversion pipeline from SpikeZIP-TF [1], ahieving the aforementional objective is non-trivial that demands to address multiple critical challenges. **Our solutions summarized as follows are composed as methodological innovations in this work.**
>
>
> **Challenge-1：As tabulated in R1-Table 1, directly applying the Quantization-Aware Training method LSQ [2] in SpikeZIP-TF on LLMs fails to efficiently achieve effective performance. Meanwhile, simply applying the vanilla PrefixQuant [3] results in that the converted SNN contains non-spike communication and introduces inefficient computation.** More specifically, PrefixQuant neglects post-q and post-softmax quantization (i.e., cannot be further converted into the spike form) as shown in Figure 3, which fails to support the conversion of matrix product ($QK^{T}$, $\text{softmax}(\frac{QK^{T}}{\sqrt{d}})V$) into spike version, thus increasing the energy consumption of converted SNNs (as tabulated in R1-Table 2).
>
>
> **Our Solution: Rather than directly applying PrefixQuant, we implement the variant PrefixQuant\*.** we insert post-q and post-softmax quantization on the basis of PrefixQuant to introduce PrefixQuant* in section 4.1. `For 4-dimensional post-softmax output, we propose a novel strategy called QK2Head-migration quantization`, as tabulated in Figure 7, R1-Table 3 and R1-Table 4, directly applying original quantization in PrefixQuant to quantize softmax output (where km=1 and qm=1) apparently degrades the performance, `while our QK2Head-migration quantization (where km=16 and qm=1) significantly improves the post-softmax quantization performance`. Ultimately, the SNN converted from the QLLM using PrefixQuant* shows better energy efficiency compared to counterpart using prefixQuant.
>
>                    R1-Table 1: Comparison between LSQ and PrefixQuant* on Llama-2-7B.
> | Model         | Quantization Method | Quantization Type         |   GPU         |  Time         | Bits         | WikiText2 Perplexity |
> |:---------------:|:----------------------------------:|:--------------:|:--------------:|:--------------:|:--------------:|:----------------------:|
> | Llama-2-7B    | LSQ  | QAT                            |          4 NVIDIA-4090                  |            6 hours                | W4A4QKVS4    | 45.28                |
> | Llama-2-7B    | PrefixQuant* | PTQ        |         1 NVIDIA-4090               |           **269 seconds**             | W4A4QKVS4    | **11.56**               |
>
>          R1-Table 2: Comparison of energy consumption between PrefixQuant, PrefixQuant* and corresponding SNNs.
>
> | Method         | Bits | post-q         |   post-softmax         |  2-7B energy(J)         | 2-7B SNNs energy(J)         | 2-13B energy(J)         | 2-13B  SNNs energy(J)  |
> |:---------------:|:----------------------------------:|:--------------:|:--------------:|:--------------:|:--------------:|:----------------------:|:----------------------:|
> | PrefixQuant    | W4A4KV4  | ✗   |          ✗                |            2.15               | 2.77(+0.62)    | 4.18                | 5.64(+1.46)              |
> | PrefixQuant*    | W4A4QKVS4  | ✓      |        ✓              |          1.64             | 0.94(-0.70)    | 3.16               |  2.08(-1.08)                |

---

> ### Author Response · Authors · 2025-11-25
> **Response to Reviewer DZKi (2/4)**
>
> **[Continue to the Answer for Weakness 1]**:
>
>      R1-Table 3: Wikitext2 perplexity with Llama-2-13B W4A4QKVS4 under various (km, qm) settings.
> | (km,qm)    | 1 | 4 | 16 | 64 |
> |:----:|:-------:|:----------:|:----------------:|:--------------:|
> | 1   | 42.31 | 31.06 | 14.77 | 13.10  |
> | 4    | 19.71 | 15.94 | 12.59 | 25.43  |
> | 16    | **7.99** | 10.02 | 22.74 | 138.62  |
> | 64   | 10.23 | 22.18 | 109.68 | 307.30 |
>
>      R1-Table 4: Wikitext2 perplexity with Llama-2-13B W4A5QKVS5 under various (km, qm) settings.
> | (km,qm)     | 1 | 4 | 16 | 64 |
> |:----:|:-------:|:----------:|:----------------:|:--------------:|
> | 1   | 8.96 | 7.45 | 6.46 | 8.88  |
> | 4    | 7.27 | 6.41 | 8.45 | 18.67  |
> | 16    | **6.28** | 8.19 | 17.08 | 105.49  |
> | 64   | 8.44 | 16.48 | 85.86 | 245.09 |
>
>
> **Challenge-2:  As tabulated in R1-Table 5, SpikeZIP-TF lacks strategies for converting LLMs-specific operators (e.g., KV Cache, RMSNorm) to their spike-equivalent version, which fails to enable the equivalence between SNNs and QLLMs.**
>
> **Our solution: We innovatively introduce Spike KV Cache (in Section 4.3), Spike SiLU, Spike RMSNorm and Spike Softmax (in Section A2).** `The introduction of spike operators above enables the equivalence between QLLMs and SNNs, which is experimentally verified in Table 4 and R1-Table 5.`
>
>                      R1-Table 5: Comparison between SpikingLLM and SpikeZIP-TF.
> |    Method   | Window Inhibition Mechanism | Spike Operators |    Bits   | Energy(J) | WikiText2 Perplexity | C4 Perplexity |
> |:-----------:|:---------------------------:|:---------------:|:---------:|:---------:|:--------------------:|:-------------:|
> | SpikeZIP-TF |           ✗                 |        ✗        | W4A4QKVS4 |    3.08   |        857.42        |     942.86    |
> |  SpikingLLM |           ✓                 |        ✓        | W4A4QKVS4 |  **0.94** |       **10.99**      |   **13.78**   |
>
> **Challenge-3: Simply applying the vanilla ST-BIF neuron (L=1 in R1-Table 6) introduces a great amount of redunant spike communication and computation, thus hampering the energy efficiency.** As ST-BIF neuron fires tenary spikes (+1,0,-1), it may existing repeatively generate +1 and -1 as neuron output (aka. spike oscillation), resulting in that the converted SNNs fail to reduce the energy consumption when converting the QLLMs to its SNN counterpart (3.16J->3.96J in R1-Table 6).
>
>
> **Our Solution: We propose a brain-inspired inihibition mechanism to mitigate such phenomenon.** As tabulated in Table 6 and R1-Table 6, with negligible performance degradation, our window inhibition mechanism effectively suppresses over-firing issues and significantly reduces the energy consumption of converted SNNs compared to corresponding QLLMs
>
>     R1-Table 6: Ablation of window inhibition mechanism on Llama-2-13B. L refers to inhibiting window length.
>
> |Method|Category| L         | Sparsity | Energy(J)         |   WikiText2 Perplexity        |  C4 Perplexity |
> |:---------------:|:---------------:|:---------------:|:----------------------------------:|:--------------:|:--------------:|:--------------:|
> | PrefixQuant* | QLLMs | -    | -  |         3.16                   |          7.99                  |            10.68                |
> | SpikingLLM | SNNs | 1    | 32.82%  |         3.96                   |          **7.72**                  |            **10.23**                |
> | SpikingLLM | SNNs | 2    | _48.94%_ |    _2.66_     |         _7.75_               |           _10.26_             |
> | SpikingLLM | SNNs | 4    | **62.51%** |   **2.08**      |         7.80              |           10.32             |
>
> **With the three solutions above, our SpikingLLM addresses the critical challenges on scaling ANN-to-SNN method up to LLMs, bridging the gap between SNNs and LLMs.**

---

> ### Author Response · Authors · 2025-11-25
> **Response to Reviewer DZKi (3/4)**
>
> **Weakness2: Building on the first point, the paper lacks a rigorous validation of the proposed method’s effectiveness and generalization. For example, in Tables 2 and 3, the authors mainly compare with PrefixQuant, without showing whether the method generalizes to other quantization approaches. Therefore, it is suggested that the authors strengthen the discussion and contribution of the ANN-to-SNN conversion method itself, rather than relying heavily on a specific quantization technique.**
>
> **[Answer for Weakness 2]**:
>
> Thanks for your insightful suggestion. We first highlight that the critical procedure of traditional ANN-to-SNN methods(e.g., SpikeZIP-TF [1], MST [4]) is to `replace the activation quantizer with equivalent spike neuron`, thus **converting the Multiply-ACcumulate operations (MACs) in to ACcumulate-Only operations (ACs).** Consequently, the activation quantization plays an important role within ANN-to-SNN (A2S) conversion pipeline. However, quantization methods that meet the following conditions are applicable to A2S.
>
> ①: **Since that the activation quantization plays an important role within ANN-to-SNN (A2S) conversion pipeline, weight-activation quantization methods are applicable to A2S**. Consequently, as tabulated in R1-Table 7, weight-only quantization method EfficientQAT [5] is not applicable to A2S.
>
> ②: **Instead of dynamic ones, static activation quantization methods are applicable to A2S**. In Section 3.2, we explain the reason for selecting static quantization method PrefixQuant rather than dynamic quantization method(e.g., OmniQuant [6]). Static quantization means that `the quantization scale of each quantizer is fixed at inference`, while `the quantization scale of each quantizer in dynamic quantization is dynamically determined by the input tensor at inference`. During SNNs inference, the threshold voltage $\vec{V}_{\rm thr}$ of neuron $\Theta$ is equal to the quantization scale $\vec{q}$ of corresponding quantizer. Consequently, `when tackling with the spiking version input(which means we can not acquire the total input before T time-step), static quantization with fixed quantization scale is more suitable to ANN-to-SNN method compared to dynamic quantization method where the quantization scale is dynamically determined by input`.
>
> ③: **Only static activation quantization methods encompassing all activations are applicable to A2S**. As illustrated in Challenge-1 of [**Answer for Weakness 1**], the vanilla PrefixQuant neglects post-q and post-softmax quantization, which fails to support the conversion of matrix product ($QK^{T}$, $\text{softmax}(\frac{QK^{T}}{\sqrt{d}})V$) into spike version, which is not applicable to A2S (as tabulated in R1-Table 7).
>
>                                    R1-Table 7: Comparison of quantization methods on A2S.
> |    Method    | weight quantization | activation quantization | activation quantization type | quantization on all activations | applicable to A2S |
> |:------------:|:-------------------:|:-----------------------:|:----------------------------:|:-------------------------------:|:-----------------:|
> | EfficientQAT |          ✓          |            ✗            |               -              |                -                |         ✗         |
> |   OmniQuant  |          ✓          |            ✓            |            dynamic           |                ✗                |         ✗         |
> |  PrefixQuant |          ✓          |            ✓            |            static            |                ✗                |         ✗         |
> | PrefixQuant* |          ✓          |            ✓            |            static            |                ✓                |         ✓         |
>
> As illustrated in solution for challenge-1 of [**Answer for Weakness 1**], we `introduce a novel QK2Head-migration post-softmax quantization alongwith post-q quantization on the basis of PrefixQuant to introduce PrefixQuant* in section 4.1`. Our PrefixQuant* meets the three conditions above, which is applicable to ANN-to-SNN conversion pipeline.

---

> ### Author Response · Authors · 2025-11-25
> **Response to Reviewer DZKi (4/4)**
>
> **[Continue to the Answer for Weakness 2]**:
>
> To verify the generalization ability of SpikingLLM, we further compare SpikingLLM with different static QLLMs on Llama-2-7B in R1-Table 8. For OmniQuant [6], we replace the dynamic activation quantizers with static activation quantizers to establish OmniQuant\*. For weight-only quantization method EfficientQAT [5], we incorporate static activation quantizers similar to PrefixQuant\* in Figure 3 to establish EfficientQAT\*. `As tabulated, our SpikingLLM enables the equivalence between the static QLLMs and SNNs, verifying the effectiveness of SpikingLLM on arbitrary static QLLMs`. We will incorporate the experimental results and detailed analysis in the revised manuscript.
>
>                  R1-Table 8: Comparison between SpikingLLM and different static QLLMs on Llama-2-7B.
> | Method            | Category        | bits    | Time-step    |  WikiText2 Perplexity    | PIQA | ARC-e | ARC-c | HellaSwag | Winogrand | Avg.  |
> |:-----------------:|:---------------------:|:----------:|:----------:|:-----------:|:-----------:|:----:|:-----:|:-----:|:---------:|:---------:|
> | OmniQuant\*  | static QLLMs  | W4A4QKVS4       |      -     |    12.49     | 70.9 | 61.2  | **35.8**  | 66.9      | **60.8**      | 59.12 |
> | SpikingLLM | SNNs  | W4A4QKVS4       |       16     |   **12.32**      | **71.1** | **61.4**  | 35.6  | **67.0**      | 60.6    | **59.14** |
> |-----------------|---------------------|----------|----------|-----------|-----------|----|-----|-----|---------|---------|-----|
> | EfficientQAT\* | static QLLMs  | W4A4QKVS4  | -    |    11.32      |     72.9      | 63.1 | **36.8**  | 69.2 | **61.8**      | 60.76 |
> | SpikingLLM | SNNs  | W4A4QKVS4       |     16     |     **10.78**      | **73.0** | **63.2**  | 36.7  | **69.3**      | **61.8**      | **60.80** |
>
>
> [**Reference**] \
> [1] K You, et al. SpikeZIP-TF: Conversion is All You Need for Transformer-based SNN. ICML 2024 \
> [2] Steven K. Esser, et al. Learned Step Size Quantization. ICLR 2020 \
> [3] M Zhao, et al. PrefixQuant: Eliminating Outliers by Prefixed Tokens for Large Language Models Quantization. \
> [4] Z Wang, et al. Masked Spiking Transformer. ICCV 2023 \
> [5] M Zhao, et al. Efficient quantization-aware training for large language models. ACL main 2025 \
> [6] W Shao, et al. Omniquant: Omnidirectionally calibrated quantization for large language models. ICLR 2024

---

> ### Author Response · Authors · 2025-11-28
> **Sincerely looking forward to the further discussions**
>
> Dear reviewer,
>
> We are wondering if our response and revision have resolved your concerns. If our response has addressed your concerns, we would highly appreciate it if you could re-evaluate our work and consider raising the score.
>
> If you have any additional questions or suggestions, we would be happy to have further discussions.
>
> Best regards,
>
> The Authors

---

### Author Response · Authors · 2025-11-29
**Response to Area Chair and All Reviewers**

Dear Area Chair and all Reviewers,

We fully understand the additional workload caused by the recent OpenReview bug and sincerely appreciate the program committee’s continued efforts to maintaining the integrity of the ICLR review process. We sincerely appreciate the dedication you and the reviewers have shown throughout the review process.

We appreciate all reviewers for their constructive and insightful feedback. The constructive feedback provided has been instrumental in refining our work with the following key clarification and improvements:

First of all, we want to highlight that `SpikingLLM is the first work that successively bridges the gap between SNN and pretrained LLM, while simutaenously achieving high output quality, high energy efficiency and no-training overhead`. Then we clarify the novelty and effectiveness of our three key contributions:

Contribution-①: In Section 4.1, we introduce PrefixQuant\* with novel QK2Head-Migration post-softmax quantization, enabling better energy efficiency compared to counterpart using PrefixQuant without post-q and post-softmax quantization (As tabulated in Table 5 and R1-Table 2).

Contribution-②: In Section 4.2, we propose a refined ST-BIF neuron with window inhibition mechanism, significantly reducing the energy consumption of converted SNNs while maintaining the performance (As tabulated in Table 6 and R1-Table 6).

Contribution-③: We innovatively introduce LLM-specific Spike Operators (Spike KV Cache in Section 4.3, Spike SiLU, Spike RMSNorm and Spike Softmax in Section A2), further enabling the equivalence between QLLMs and SNNs, which is experimentally verified in Table 4 and R1-Table 5.

Furthermore, we provide extended evaluation and analysis in response to the reviewers' comments:

①: Experimentall results in R2-Table 3 that demonstrate the generalization ability of our SpikingLLM on arbitrary quantized LLMs.

②: Comprehensive comparison with other efficient LLMs (e.g., BitNet-1.58b, MatMul-free LLM etc.) in Table A4, Table A5 and R2-Table 4 that validate the effectiveness of our SpikingLLM.

③: Experimentally demonstrating the adaptability of our method beyond language modeling and reasoning in R4-Table 5.

With the clarification above and responses below, we kindly request your assistance in facilitating a re-evaluation of our work after our rebuttal. Your support is crucial to us, and we genuinely appreciate your attention to this matter. lf you have any questions, we are happy to provide further details. Thanks again for your time and dedication to the community.

Best regards,

The Authors

---

### Meta-Review · Area_Chair_P7LT · 2026-01-01

**Summary:**

The reviewers generally acknowledge the paper’s ambitious goal of bridging spiking neural networks (SNNs) with large language models (LLMs) for energy-efficient inference and recognize the technical effort and extensive experimental validation presented. However, the consensus among the reviewers is that the novelty and contribution of the work are insufficient for acceptance at ICLR 2026, falling marginally below the acceptance threshold. The primary concerns are as follows:

Limited Novelty and Incremental Contribution: Multiple reviewers (DZKI, HRJH, mvGP) noted that the core methodology heavily relies on established techniques. The ANN-to-SNN conversion pipeline is deemed conventional, and the proposed PrefixQuant* is viewed largely as an adaptation of the existing PrefixQuant method. The integration of ternary spiking operators is seen as an extension of prior work like SpikeZIP-TF. The reviewers felt that the performance gains are attributed more to the underlying quantization method (PrefixQuant) rather than to fundamental methodological innovations from the authors.

Lack of Problem Definition and Generalization: Reviewers (DZKI, mvGP) argued that the paper does not clearly define a new problem within either the SNN or quantization domains. The validation was perceived as overly reliant on a single quantization method (PrefixQuant), without sufficiently demonstrating the generality of the proposed SpikingLLM framework across other quantization approaches. Although the authors provided additional experiments in the rebuttal, the initial presentation weakened the case for broad applicability.

Insufficient Analysis of Critical Design Choices: Several weaknesses highlighted a lack of in-depth analysis:

Window Inhibition Mechanism (4kKu, HRJH): Concerns were raised about potential distortion of activation magnitudes and the impact of changing spike temporal distributions. The equivalence between the modified neuron and the quantizer was questioned, despite the authors' explanations and experimental MSE data.

Approximation Error Accumulation (4kKu): The paper did not adequately analyze how the approximation errors from converting non-spiking operators (e.g., Softmax, KV Cache) propagate through deep transformer layers, which is critical for assessing fidelity.

Latency and Practical Deployment (4kKu, HRJH): The multi-timestep inference requirement (16-32 steps) introduces significant latency, which is a major practical drawback for LLM deployment. The rebuttal argued for energy efficiency on neuromorphic hardware, but reviewers remained concerned about the real-time trade-offs and suitability for large-scale, latency-sensitive applications.

Presentation and Technical Depth: While the presentation was generally clear, the technical pipeline was considered dense and difficult for non-specialists (4kKu). The relationship between quantization and SNN conversion, though addressed in the rebuttal, was initially deemed insufficiently explored (HRJH).

Rebuttal Response: The authors provided a detailed and comprehensive rebuttal, addressing each point with additional experiments, clarifications, and extended analyses. They demonstrated the generalization of their method to other static quantization techniques, provided layer-by-layer error analysis, and further justified the window inhibition mechanism and energy-latency trade-offs.

Final Assessment: Despite the thorough rebuttal, the reviewers' fundamental concerns regarding the incremental nature of the contribution and the remaining questions about practical deployment impact were not fully alleviated. The work is seen as a solid engineering effort within an established technical paradigm but does not meet the bar for a significant conceptual or methodological advance expected at ICLR. Therefore, the recommendation is to reject the submission.

**Reviewer Concerns:**

Effectively Addressed in Rebuttal:

Generalization: Demonstrated applicability to other static quantization methods via OmniQuant* and EfficientQAT*.

Window Inhibition Analysis: Provided quantitative MSE data showing minimal impact on activations.

Error Accumulation: Showed layer-by-layer error remains very low (~10^-5) in deep models.

Comparisons: Added extensive comparisons with SpikeZIP-TF, SpikeLLM, and BitNet, highlighting empirical advantages.

Task Generalization: Supplementary results on factual knowledge tasks (SciQ, BoolQ) confirmed adaptability.

Outstanding Fundamental Concerns:

Limited Novelty: Reviewers maintained that the work represents a solid engineering integration and optimization within the established ANN-to-SNN conversion paradigm, but lacks the conceptual breakthrough expected for ICLR. This is the primary reason for rejection.

Inference Latency: The requirement for 16-32 timesteps inherently multiplies inference latency. While suitable for energy-critical edge scenarios, this remains a significant practical limitation for broad LLM deployment, and the trade-off was not resolved.

**Reviewer Scores:**

DZKI (Initial: 4)
Likely unchanged. While technical responses were solid, the fundamental concern about limited novelty may persist.

mvGP (Initial: 4)
Likely unchanged or change to 6.  Some specific technical concerns (choice of quantizer, comparisons) were addressed.

HRJH (Initial: 4)
Likely unchanged. The multi-timestep latency issue represents a core trade-off that the authors’ justification (edge efficiency) does not resolve.

4kKu (Initial: 4)
Likely unchanged or change to 6. Some requests for quantitative analysis (window inhibition, error accumulation, latency curves) were satisfied.

Overall Outlook
Discussion could have led to a split decision. Given ICLR’s high bar and the initial consensus on “marginally below acceptance,” the paper should still be rejected due to insufficient perceived novelty and the inherent latency trade-off.

---

### Decision · Program_Chairs · 2026-01-26

Reject